

# The $\mathcal{W}$-algebra bootstrap of 6d $\mathcal{N} = (2, 0)$ theories

**Mitchell Woolley**

Centre for Theoretical Physics, Department of Physics and Astronomy,
Queen Mary University of London, London E1 4NS, UK

mitchell.woolley@qmul.ac.uk

## Abstract

6d (2,0) SCFTs of type $\mathfrak{g}$ have protected subsectors that were conjectured in [1] to be captured by $\mathcal{W}_{\mathfrak{g}}$ algebras. We write down the crossing equations for mixed four-point functions $\langle S_{k_1} S_{k_2} S_{k_3} S_{k_4} \rangle$ of 1/2-BPS operators $S_{k_i}$ in 6d (2,0) theories and detail how a certain twist reduces this system to a 2d meromorphic CFT multi-correlator bootstrap problem. We identify the relevant 6d (2,0) $\mathcal{W}$-algebras of type $\mathfrak{g} = \{A_{N-1}, D_N\}$ as truncations of $\mathcal{W}_{1+\infty}$ and solve OPE associativity conditions for their structure constants, both using OPEdefs and the holomorphic bootstrap of [2]. With this, we solve the multi-correlator bootstrap for twisted 6d four-point correlators $\mathcal{F}_{k_1 k_2 k_3 k_4}$ involving all $S_{k_i}$ up to $\{k_i\} = 4$ and extract closed-form expressions for 6d OPE coefficients. We describe the implications of our CFT data on conformal Regge trajectories of the (2,0) theories and finally, demonstrate the consistency of our results with protected higher-derivative corrections to graviton scattering in M-theory on $AdS_7 \times S^4/\mathbb{Z}_{\mathfrak{o}}$.

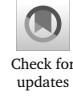

# 1  Introduction

The 6d $\mathcal{N} = (2,0)$ superconformal field theories (SCFTs) hold a distinguished place in the space of quantum field theories (QFTs), both as non-Lagrangian, strongly-interacting SCFTs of maximal dimension and as a window into quantum aspects of M-theory. The existence of 6d (2,0) theories was deduced in [3] from a decoupling limit of type IIB string theory on ADE orbifold singularities, which implied their classification by a simply-laced Lie algebra $\mathfrak{g} = \{A_{N-1}, D_N, E_6, E_7, E_8\}$. The $A_{N-1}$ theories were recognized in [4] as the infrared limit of the worldvolume theory of $N$ coincident M5 branes which, along with the $D_N$ theories were identified as early examples of the AdS/CFT correspondence with duality to M-theory on $AdS_7 \times S^4/\mathbb{Z}_{\mathfrak{o}}$ for $\mathfrak{o} = 1, 2$ [5–7]. Since then, the (2,0) theories have maintained their prominence and mystique by rationalizing a vast web of dualities among supersymmetric QFTs in lower dimensions, while remaining inaccessible directly via traditional field theory techniques.

Lacking a Lagrangian formulation, the 6d (2,0) theories are studied naturally through the lens of the conformal bootstrap, which views conformal field theories (CFTs) abstractly as an algebra of local operators parameterized by a set of CFT data. In analogy to the discovery in 4d $\mathcal{N} = 2$ SCFTs in [8], the authors of [1] demonstrated that the algebra of 6d (2,0) operators admits a closed subalgebra isomorphic to that of a 2d chiral algebra. It was conjectured that the 2d chiral algebra of a 6d (2,0) theory of type $\mathfrak{g}$ is in fact a $\mathcal{W}_{\mathfrak{g}}$ algebra with a central charge given by the 6d anomaly polynomial $c_{\mathfrak{g}}$. This construction implies that for these special protected 6d operators, we inherit an infinite set of CFT data from these solveable, yet non-trivial 2d meromorphic CFTs.

This discovery jump-started the numerical superconformal bootstrap program for 6d $(2,0)$ theories using the linear and semi-definite programming strategies born out of the revival of the conformal bootstrap [9, 10]. The authors of [11] initiated this for four-point functions $\langle S_2 S_2 S_2 S_2 \rangle$ of 1/2-BPS superconformal primaries $S_2$ of the 6d stress tensor multiplet.[1] Without additional input, numerical bootstrap constraints on $\langle S_2 S_2 S_2 S_2 \rangle$ apply universally to any 6d (2,0) theory with a stress tensor. Moreover, the protected $\mathcal{W}$-algebra type CFT data entering in this correlator is restricted to the universal Virasoro sector and could have even been

---

[1] The correlator $\langle S_2 S_2 S_2 S_2 \rangle$ was later revisited in [12] using semidefinite programming with higher precision as well as using alternative conceptual and numerical strategies in [13, 14].

computed without a chiral algebra interpretation. This situation changes when we consider more general four-point functions of 1/2-BPS operators $\langle S_{k_1} S_{k_2} S_{k_3} S_{k_4} \rangle$, where the spectrum of $S_{k_i}$ will depend on $\mathfrak{g}$ and the infinite set of protected CFT data provided by $\mathcal{W}_{\mathfrak{g}}$ are non-trivial functions of 6d central charge $c_{\mathfrak{g}}$. The latter data could either be compared with numerical bounds or inputted into the bootstrap to obtain sharper bounds on the remaining data.

The aim of this work is to compute explicit 6d (2,0) CFT data using meromorphic CFT and the vertex operator algebra (VOA) bootstrap. We achieve this by studying the system of eight non-trivial four-point functions of 1/2-BPS operators $\langle S_{k_1} S_{k_2} S_{k_3} S_{k_4} \rangle$ up to and including $\{k_i\} = 4$. Along the way, we outline the structure of the multi-correlator bootstrap for (2,0) theories and address several technicalities. We also elaborate on the $\mathcal{W}_{\mathfrak{g}}$-algebra conjecture for the $\mathfrak{g} = D_N$ family of (2,0) theories and use this to demonstrate the consistency of the $\mathcal{W}$-algebra conjecture and perturbative M-theory for both families of holographic 6d (2,0) theories.

To this end, our paper is organized as follows. In Section 2, we review the form of four-point correlation functions $\mathcal{G}_{k_1 k_2 k_3 k_4}$ of non-identical 1/2-BPS operators $S_{k_i}$ in 6d (2,0) theories and their (super)conformal block decompositions. We describe the consequences of $\mathfrak{osp}(8^*|4)$ selection rules and physical considerations on the $S_p \times S_q$ OPE and use crossing symmetry to formulate the mixed correlator bootstrap equations.

In Section 3, we demonstrate the twist that reduces the superconformal Ward identities to a holomorphicity constraint on the twisted correlator $\mathcal{F}_{k_1 k_2 k_3 k_4}$. We list the 6d operators that survive this procedure and explicitly twist an arbitrary mixed correlator superblock decomposition, presenting useful technical results along the way. The result is a global $SL(2, \mathbb{R})$ block decomposition involving 6d CFT data that can be directly compared with that of a chiral algebra. We briefly describe an ambiguity in such a comparison, known as the filtration problem.

In Section 4, we elaborate on the particular $\mathcal{W}_{\mathfrak{g}}$ algebras that were conjectured in [1] to be associated with 6d (2,0) theories of type $\mathfrak{g}$. We outline the VOA bootstrap method needed to compute OPE data beyond the universal Virasoro sector and demonstrate how both 6d (2,0) $\mathcal{W}_{\mathfrak{g}}$ algebras of type $A_{N-1}$ and $D_N$ arise naturally from a particular parameter choice of the truncations of $\mathcal{W}_{1+\infty}$. Our discussion of $\mathcal{W}_{D_N}$ includes the explicit CFT datum $C_{44}^4(D_N)$ and Pfaffian operator that, via the 6d (2,0) theory, will encode quantum properties of $N$ M5 branes on a $\mathbb{Z}_2$ orbifold.

In Section 5, we use the holomorphic bootstrap method of [2] to reformulate $\mathcal{W}_{\mathfrak{g}}$ algebra four-point correlators into closed-form expressions involving minimal low-lying data. We compare this with our twisted 6d superblock expansion and extract 6d CFT data. Our data suggests generalizations of the conformal Regge trajectories discussed in [13] and we use it to make predictions for trajectories in $\langle S_3 S_3 S_p S_p \rangle$.

In Section 6, we make contact with 6d (2,0) theories' holographic relation to M-theory. After describing perturbative supergraviton scattering in M-theory on $\mathbb{R}^{11}$, we recall how compactification on $AdS_7 \times S^4/\mathbb{Z}_\mathfrak{o}$ recasts this EFT expansion in terms of the 6d (2,0) central charge $c_{\mathfrak{g}}$. With this, we demonstrate the consistency of the large-$c_{\mathfrak{g}}$ expansion of all $\mathcal{W}_{\mathfrak{g}}$ type 6d data with protected higher-derivative terms in the effective action on $AdS_7$.

Several technical results are relegated to Appendices and an ancillary `Mathematica` file. Our conventions for 6d superblocks are reviewed in Appendix A, while Appendix B includes all of the VOA bootstrap relations needed in the main text, plus results for the exceptional 6d (2,0) theories of type $\mathfrak{g} = E_6, E_7, E_8$, obtained from the Jacobi identites $(W_i W_j W_k)$ for low $i, j, k > 2$, as explained in Section 4. Our ancillary `Mathematica` file presents all twisted correlators $\mathcal{F}_{k_1 k_2 k_3 k_4}$ up to $\{k_i\} = 4$, the 6d data we computed therefrom, and more VOA bootstrap relations.

## 2 Mixed four-point functions in 6d (2,0) theories

We begin by reviewing the mixed four-point functions of 1/2-BPS superconformal primaries that are central to this work. After describing the constraints that the 6d $\mathcal{N} = (2,0)$ superconformal algebra $\mathfrak{osp}(8^*|4)$ imposes on these correlators, we interpret the selection rules and use crossing symmetry to write down the mixed correlator bootstrap equations.

### 2.1 Four-point functions of 1/2–BPS operators

We will be concerned with four-point functions of 1/2-BPS superconformal primary operators $S_k$. These are $\mathfrak{so}(6)$ scalars with $\Delta = 2k$ and transform in the $[k, 0]$ of $\mathfrak{so}(5)_R$, where $k \in \mathbb{N}$. The operator $S_1$ is the free scalar accounting for the center of mass degree of freedom of the M5 branes and decouples from the interacting theories we will study. The lowest dimension 1/2-BPS scalar in interacting theories is $S_2$, which is the superconformal primary of the stress tensor multiplet with $\Delta = 4$. Unlike $S_k$ for $k > 2$, $S_2$ exists universally in all local 6d (2,0) theories. In this work, we will consider mixed four-point functions involving $S_k$ with $2 \leq k \leq 4$, although $S_3$ will be absent from $D_{N>3}$ theories. We can treat $S_k$ as a rank-$k$ traceless symmetric product of the $[1, 0]$ of $\mathfrak{so}(5)_R$, which we denote by $S_{I_1 \dots I_k}(x)$, where $I_i = 1, \dots 5$. As is standard in the superconformal bootstrap, we will contract $S_{I_1 \dots I_k}(x)$ with auxiliary complex $SO(5)_R$ polarization vectors $t^I$:

$$S_k(x, t) \equiv S_{I_1 \dots I_k}(x) t^{I_1} \cdots t^{I_k}. \tag{1}$$

This transfers $\mathfrak{so}(5)_R$ representation theoretical information into polynomials of products of $t^{I_i}$, where tracelessness is encoded by the null condition $t^I t_I = 0$. Constraints from $\mathfrak{osp}(8^*|4)$ superconformal symmetry allow us to factorize four-point functions of 1/2-BPS operators into a kinematic factor and a dynamical function $\mathcal{G}_{k_1 k_2 k_3 k_4}$. We adapt the 3d $\mathcal{N} = 8$ and 4d $\mathcal{N} = 4$ conventions of [15] and [16] to 6d $\mathcal{N} = (2,0)$ by writing our four-point function as[2]

$$\langle S_{k_1}(x_1, t_1) S_{k_2}(x_2, t_2) S_{k_3}(x_3, t_3) S_{k_4}(x_4, t_4) \rangle$$
$$= \left( \frac{t_{12}}{x_{12}^4} \right)^{\frac{k_1+k_2}{2}} \left( \frac{t_{34}}{x_{34}^4} \right)^{\frac{k_3+k_4}{2}} \left( \frac{t_{14}}{t_{24}} \right)^{\frac{k_{12}}{2}} \left( \frac{t_{12} t_{34}}{t_{14} t_{24}} \right)^{\frac{k_{34}}{2}} \left( \frac{x_{14}^4 x_{24}^4}{x_{12}^4 x_{34}^4} \right)^{\frac{k_{34}}{2}} \mathcal{G}_{k_1 k_2 k_3 k_4}(U, V; \sigma, \tau), \tag{2}$$

where we consider operator orderings with $k_1, k_2, k_3 \leq k_4$ and we define $x_{ij} = x_i - x_j$, $t_{ij} = t_i \cdot t_j$, and $k_{ij} = k_i - k_j$. The dynamical function $\mathcal{G}_{k_1 k_2 k_3 k_4}$ depends on conformal cross-ratios $U$ and $V$ and $\mathfrak{so}(5)_R$ cross-ratios $\sigma$ and $\tau$ defined by

$$U = \frac{x_{12}^2 x_{34}^2}{x_{13}^2 x_{24}^2}, \qquad V = \frac{x_{14}^2 x_{23}^2}{x_{13}^2 x_{24}^2}, \qquad \sigma = \frac{t_{13} t_{24}}{t_{12} t_{34}}, \qquad \tau = \frac{t_{14} t_{23}}{t_{12} t_{34}}. \tag{3}$$

This organization implies that $\mathcal{G}_{k_1 k_2 k_3 k_4}$ is a polynomial in $\sigma, \tau$ of degree $\min\{k_i\}$, modulo a factor $\tau^{\frac{1}{2}(k_2+k_3-k_1-k_4)}$ that we include for kinematical configurations satisfying $k_1 + k_4 < k_2 + k_3$. The latter subtlety will become important when we study the correlator $\langle S_2 S_4 S_4 S_4 \rangle$.

The bosonic subalgebra of $\mathfrak{osp}(8^*|4)$ allows us to decompose $\mathcal{G}_{k_1 k_2 k_3 k_4}$ into a basis of exchanged $\mathfrak{so}(5)_R$ representations. The allowed irreducible representations (irreps) exchanged in the OPE $S_{k_1} \times S_{k_2}$ are given by the $\mathfrak{so}(5)_R$ tensor product

$$[k_1, 0] \otimes [k_2, 0] = \bigoplus_{m=0}^{\min\{k_1, k_2\}} \bigoplus_{n=0}^{\min\{k_1, k_2\}-m} [k_1 + k_2 - 2(m+n), 2n]. \tag{4}$$

---

[2]Our kinematic factor does not vary between "Case I" and "Case II" kinematics, as defined in e.g. [17, 18]. Instead, we will later compensate for negative powers of $t_{14}$ in "Case II" by including appropriate powers of $\tau$ in $\mathcal{G}_{k_1 k_2 k_3 k_4}$, following the 4d $\mathcal{N} = 4$ analogy in [19].

Focusing on the $s$-channel, the admissible $\mathfrak{so}(5)_R$ irreps in $\mathcal{G}_{k_1 k_2 k_3 k_4}$ are captured by $([k_1,0] \otimes [k_2,0]) \cap ([k_3,0] \otimes [k_4,0])$. We can decompose $\mathcal{G}_{k_1 k_2 k_3 k_4}$ into these representations by writing

$$\mathcal{G}_{k_1 k_2 k_3 k_4}(U,V;\sigma,\tau) = \tau^\delta \sum_{m=\frac{k_1+k_2-2\min\{k_i\}}{2}}^{\frac{k_1+k_2}{2}} \sum_{n=\frac{k_1+k_2-2\min\{k_i\}}{2}}^{m} Y_{mn}^{\frac{|k_{12}-k_{34}|}{2},\frac{|k_{23}-k_{14}|}{2}}(\sigma,\tau) A_{mn}^{k_{12},k_{34}}(U,V). \quad (5)$$

The functions $Y_{mn}^{a,b}(\sigma,\tau)$ are eigenfunctions of the $\mathfrak{so}(5)_R$ quadratic Casimir associated with exchanged irreps $[2n,2(m-n)]$. They are polynomials in $\sigma$ and $\tau$ of degree $m - \frac{a+b}{2}$ and can be computed as reviewed in Appendix A. The exponent $\delta$ is

$$\delta = \begin{cases} 0, & k_1 + k_4 \geq k_2 + k_3, \\ \frac{k_2+k_3-k_1-k_4}{2}, & k_1 + k_4 < k_2 + k_3, \end{cases} \quad (6)$$

and serves to cancel negative powers of $t_{14}$ in the kinematic factor, as described in [19]. We can then take the $s$-channel OPEs $S_{k_1} \times S_{k_2}$ and $S_{k_3} \times S_{k_4}$ and expand $A_{mn}^{k_{12},k_{34}}(U,V)$ in 6d bosonic conformal blocks $G_{\Delta,\ell}^{\Delta_{12},\Delta_{34}}(U,V)$ as

$$A_{mn}^{k_{12},k_{34}}(U,V) = U^{k_{34}} \sum_{\Delta,\ell} \lambda_{k_1 k_2 \mathcal{O}_{\Delta,\ell,mn}} \lambda_{k_3 k_4 \mathcal{O}_{\Delta,\ell,mn}} G_{\Delta,\ell}^{\Delta_{12},\Delta_{34}}(U,V). \quad (7)$$

The fermionic subalgebra of $\mathfrak{osp}(8^*|4)$ imposes additional constraints which are captured by the superconformal Ward identities [20]

$$(z\partial_z - 2\alpha\partial_\alpha)\mathcal{G}_{k_1 k_2 k_3 k_4}(z,\bar{z};\alpha,\bar{\alpha})\Big|_{\alpha \to z^{-1}} = (\bar{z}\partial_{\bar{z}} - 2\bar{\alpha}\partial_{\bar{\alpha}})\mathcal{G}_{k_1 k_2 k_3 k_4}(z,\bar{z};\alpha,\bar{\alpha})\Big|_{\bar{\alpha} \to \bar{z}^{-1}} = 0, \quad (8)$$

where the variables $z,\bar{z}$ and $\alpha,\bar{\alpha}$ can be written in terms of conformal and R-symmetry cross-ratios as

$$U = z\bar{z}, \qquad V = (1-z)(1-\bar{z}), \qquad \sigma = \alpha\bar{\alpha}, \qquad \tau = (1-\alpha)(1-\bar{\alpha}). \quad (9)$$

These constraints will have two important consequences for this work, with the first being a more practical one. Conformal symmetry allowed us to decompose $\mathcal{G}_{k_1 k_2 k_3 k_4}$ in a sum of conformal blocks $G_{\Delta,\ell}^{\Delta_{12},\Delta_{34}}(U,V)$, i.e. in a basis of functions that individually satisfy the $\mathfrak{so}(6,2)$ Casimir equation. Superconformal symmetry allows us to repackage $\mathcal{G}_{k_1 k_2 k_3 k_4}$ into an expansion in superblocks $\mathfrak{G}_{\mathcal{X}}^{\Delta_{12},\Delta_{34}}(U,V;\sigma,\tau)$, i.e. a basis of functions that individually satisfy (8). So we can write

$$\mathcal{G}_{k_1 k_2 k_3 k_4}(U,V;\sigma,\tau) = \sum_{\mathcal{X} \in (S_1 \times S_2) \cap (S_3 \times S_4)} \lambda_{k_1 k_2 \mathcal{X}} \lambda_{k_3 k_4 \mathcal{X}} \mathfrak{G}_{\mathcal{X}}^{\Delta_{12},\Delta_{34}}(U,V;\sigma,\tau), \quad (10)$$

where superblocks then take the form

$$\mathfrak{G}_{\mathcal{X}}^{\Delta_{12},\Delta_{34}}(U,V;\sigma,\tau) \quad (11)$$

$$= \tau^\delta U^{k_{34}} \sum_{m=\frac{k_1+k_2-2\min\{k_i\}}{2}}^{\frac{k_1+k_2}{2}} \sum_{n=\frac{k_1+k_2-2\min\{k_i\}}{2}}^{m} Y_{mn}^{\frac{|k_{12}-k_{34}|}{2},\frac{|k_{23}-k_{14}|}{2}}(\sigma,\tau) \sum_{\mathcal{O} \in \mathcal{X}} \mathcal{C}_{\mathcal{O}_{\Delta,\ell,mn}}^{\mathcal{X},\Delta_{12},\Delta_{34}} G_{\Delta,\ell}^{\Delta_{12},\Delta_{34}}(U,V).$$

In passing from the the block decomposition in (5) and (7) to the superblock decomposition in (10), supersymmetry has identified all products of three-point coefficients

of superdescendant primaries $\mathcal{O}_{\Delta,\ell,mn}$ in a supermultiplet to be proportional to those of the superconformal primary, which we also denote $\mathcal{X}$. The proportionality constants $\mathcal{C}^{\mathcal{X},\Delta_{12},\Delta_{34}}_{\mathcal{O}_{\Delta,\ell,mn}} = \frac{\lambda_{k_1 k_2 \mathcal{O}_{\Delta,\ell,mn}} \lambda_{k_3 k_4 \mathcal{O}_{\Delta,\ell,mn}}}{\lambda_{k_1 k_2 \mathcal{X}} \lambda_{k_3 k_4 \mathcal{X}}}$ weight different primaries in a supermultiplet and can be computed by expanding the ansatz (11) around $z, \bar{z} \ll 1$ (e.g. using Jack polynomials) and imposing (8) order-by-order, following the strategy in [21–23]. To actually generate the list of superdescendant primaries $\mathcal{O}_{\Delta,\ell,mn}$ in a given supermultiplet $\mathcal{X}$, we used the Python implementation [24] of the Racah-Speiser algorithm [25, 26], taking into account the R-symmetry selection rules and the $\ell$ selection rules discussed in the following subsection.

## 2.2 Selection rules and the OPE

The superconformal multiplets $\mathcal{X}$ that are allowed to appear in a given OPE $S_{k_1} \times S_{k_2}$ are dictated by the selection rules for 1/2-BPS operators worked out in [27, 28] and summarized in [11]. Operators $S_k$ are superconformal primaries of the 1/2-BPS supermultiplets $\mathcal{D}[k, 0]$, which have the following tensor product

$$
\begin{aligned}
\mathcal{D}[k_1, 0] \otimes \mathcal{D}[k_2, 0] = &\bigoplus_{i=0}^{\min\{k_1,k_2\}} \bigoplus_{j=0}^{\min\{k_1,k_2\}-i} \mathcal{D}[k_1 + k_2 - 2(i+j), 2j] \\
&\oplus \bigoplus_{i=1}^{\min\{k_1,k_2\}} \bigoplus_{j=0}^{\min\{k_1,k_2\}-i} \bigoplus_{l=0}^{\infty} \mathcal{B}[k_1 + k_2 - 2(i+j), 2j]_l \\
&\oplus \bigoplus_{i=2}^{\min\{k_1,k_2\}} \bigoplus_{j=0}^{\min\{k_1,k_2\}-i} \bigoplus_{l=0}^{\infty} \bigoplus_{\Delta>6+l+2(k_1+k_2-2i)}^{\infty} \mathcal{L}[k_1 + k_2 - 2(i+j), 2j]_{\Delta,l}.
\end{aligned}
\tag{12}
$$

A superconformal multiplet $\mathcal{X}[d_1, d_2]_{\Delta,\ell}$ is defined to have a superconformal primary that transforms in the $[d_1, d_2]$ of $\mathfrak{so}(5)_R$ R-symmetry and in the rank-$\ell$ traceless symmetric irrep $[0, \ell, 0]$ of the Lorentz algebra $\mathfrak{su}(4)$. The letter $\mathcal{X}$ specifies which shortening condition (or lack thereof) that the superconformal primary obeys. The shortening conditions of supermultiplets appearing in (12) are[3]

$$
\begin{aligned}
\mathcal{D}[p, q]: &\quad \Delta = 2(p+q), \\
\mathcal{B}[p, q]_\ell: &\quad \Delta = 2(p+q) + 4 + \ell, \\
\mathcal{L}[p, q]_{\Delta,\ell}: &\quad \Delta > 2(p+q) + 6 + \ell.
\end{aligned}
\tag{13}
$$

While selection rules enumerate all of the exchanged supermultiplets allowed by representation theory, additional considerations will constrain what appears in the physical OPE. For example, when $k_1 = k_2$, Bose symmetry dictates that $\mathfrak{so}(5)_R$ irreps $[2a, 2b]$ must have even/odd $\ell$ for even/odd $b$ (corresponding to a symmetric/antisymmetric irrep). The case $k_1 = k_2$ would also nominally give rise to $\mathcal{B}[0, 0]_\ell$ exchanges, however these are higher-spin conserved currents which do not exist in interacting theories [29]. The $S_k \times S_k$ self-OPE will always include the stress tensor superprimary $S_2 \in \mathcal{D}[2, 0]$. In terms of the $c$-coefficient of the canonically normalized stress tensor two-point function

$$
\langle T_{\mu\nu}(x) T_{\rho\sigma}(0) \rangle = c \frac{84}{\pi^6} \frac{\mathcal{I}_{\mu\nu\rho\sigma}(x)}{|x|^{12}},
\tag{14}
$$

the $S_2$ exchanged in a correlator $\langle S_{k_1} S_{k_1} S_{k_2} S_{k_2} \rangle$ yields the three-point coefficient product

$$
\lambda_{k_1 k_1 \mathcal{D}[2,0]} \lambda_{k_2 k_2 \mathcal{D}[2,0]} = \frac{k_1 k_2}{c},
\tag{15}
$$

---

[3]See [25, 26] for a more complete treatment of 6d $\mathcal{N} = (2,0)$ superconformal representation theory.

which follows from the free theory with $c = 1$. In this normalization, the $c$-coefficients of known the (2,0) theories labeled by $\mathfrak{g} \in \{A_{N-1}, D_N, E_6, E_7, E_8\}$ are given by [1]

$$c_{\mathfrak{g}} = 4d_{\mathfrak{g}}h_{\mathfrak{g}}^{\vee} + r_{\mathfrak{g}}, \tag{16}$$

where $d_{\mathfrak{g}}, h_{\mathfrak{g}}^{\vee}$, and $r_{\mathfrak{g}}$ are the dimension, dual Coxeter number, and rank of $\mathfrak{g}$, respectively. For brevity, we while often suppress the $\mathfrak{g}$ subscript when $\mathfrak{g}$ is implied by the context.

Representation theory also does not specify which explicit operators play the role of a given superprimary $\mathcal{X}[d_1, d_2]_{\Delta, \ell}$. For instance, operators $S_k$ are merely examples of superconformal primaries of the 1/2-BPS multiplets $\mathcal{D}[k, 0]$ and we can also construct 1/2-BPS superprimaries out of linear combinations of normal-ordered products (i.e. multi-traces[4]) of $S_k$. Lacking an $S_1$ operator, the superprimaries $\mathcal{D}[2, 0]$ and $\mathcal{D}[3, 0]$ appearing in this work are unambiguously $S_2$ and $S_3$. However, $\mathcal{D}[4, 0]$ can be an admixture of $S_4$ and the $[4, 0]$ projection of $: S_2 S_2 :$. While we will not consider correlators involving external composite operators, they are necessarily exchanged in the OPE of any 1/2-BPS operators $S_i \times S_j$ and therefore play a role in every correlator we consider. Additional considerations are needed to specify the single- and multi-trace superprimaries appearing in a given OPE.

One such set of considerations come from holography, wherein single-trace and multi-trace realizations of $\mathcal{D}[k, 0]$ can be partially distinguished by their appearance in a large-$c_{\mathfrak{g}}$ expansion. For instance, when $S_{k_1} \times S_{k_2}$ exchanges a multiplet $\mathcal{D}[p, 0]$ with $p < k_1 + k_2$, the leading contribution to the three-point coefficient $\lambda_{k_1 k_2 \mathcal{D}[p, 0]}$ is $O(c^{1/2})$ and comes from the single-trace contribution $S_p$. On the other hand, finiteness of the effective action on the $AdS_7$ dual requires that the exchanged $\mathcal{D}[k_1 + k_2, 0]$ has no single-trace contribution, as argued in [30, 31] and reviewed in [32]. The latter extremal 3-point coefficient $\lambda_{k_1 k_2 \mathcal{D}[k_1 + k_2, 0]}$ is $O(1)$, stemming from its $: S_{k_1} S_{k_2} :$ double-trace contribution that survives in the generalized free field (GFF) limit $c \to \infty$.

Additional considerations include that, just as we excluded the exchange of $\mathcal{B}[0, 0]_{\ell}$ multiplets with superprimary $: S_1 \partial^{\ell} S_1 :$, we also discard any other multi-trace operators involving $S_1$ in interacting theories. This will generically rule out certain low-lying $\mathcal{D}$ and $\mathcal{B}$ multiplets from (12). Finally, we note that 6d (2,0) theories of type $D_N$ do not contain any $\mathcal{D}[k, 0]$ with odd $k$ except the Pfaffian operator $P_N$ which is the superconformal primary of $\mathcal{D}[N, 0]$, where $N$ can be odd. We will see in Section 4 that the known OPEs of the corresponding $\mathcal{W}$-algebras will independently support all of the above statements. As an example of these ideas in practice, consider the superselection rule for $S_2 \times S_3$:

$$
\mathcal{D}[2, 0] \otimes \mathcal{D}[3, 0] = \underbrace{\mathcal{D}[1, 0]}_{S_1} \oplus \underbrace{\mathcal{D}[3, 0]}_{S_3} \oplus \underbrace{\mathcal{D}[1, 2]}_{:S_1 S_2: |_{[1,2]}} \oplus \underbrace{\mathcal{D}[5, 0]}_{:S_2 S_3: |_{[5,0]}} \oplus \underbrace{\mathcal{D}[3, 2]}_{:S_2 S_3: |_{[3,2]}} \oplus \underbrace{\mathcal{D}[1, 4]}_{:S_2 S_3: |_{[1,4]}}
$$
$$
\oplus \bigoplus_{\ell = 0, 1, 2, \dots}^{\infty} \left( \underbrace{\mathcal{B}[1, 0]_{\ell}}_{:S_1 \partial^{\ell} S_3: |_{[1,0]}} \oplus \underbrace{\mathcal{B}[3, 0]_{\ell}}_{:S_2 \partial^{\ell} S_3: |_{[3,0]}} \oplus \underbrace{\mathcal{B}[1, 2]_{\ell}}_{:S_2 \partial^{\ell} S_3: |_{[1,2]}} \right) \tag{17}
$$
$$
\oplus \bigoplus_{\ell = 0, 1, 2, \dots}^{\infty} \bigoplus_{\Delta > 8 + \ell}^{\infty} \mathcal{L}[1, 0]_{\Delta, \ell}.
$$

As discussed above, we removed $S_1$ and composite operators thereof. We also exclude $S_5$ and identify $\mathcal{D}[p_1, p_2]$ with $p_1 + p_2 = 5$ with different projections of $: S_2 S_3 :$ and identify the semi-short operators $\mathcal{B}[p_1, p_2]_{\ell}$ with $p_1 + p_2 = 3$ as projections of $: S_2 \partial^{\ell} S_3 :$. The $\mathcal{L}[1, 0]_{\Delta, \ell}$ multiplets are unprotected and therefore have no general structure *a priori*. At large $c$, the leading

---

[4]Lacking a Lagrangian or even a conventional gauge theoretic interpretation, calling 6d (2,0) operators "single-trace" or "multi-trace" is an abuse of notation that we will continue in this work.

contributions involve double-trace operators of the schematic form $: S_p \partial^\ell \square^n S_{p+1} :|_{[1,0]}$. At fixed bare twist $t^{(0)} = 4p + 2 + 2n \geq 10$, there is mixing between $\left\lfloor \frac{t^{(0)}-6}{4} \right\rfloor$ such degenerate operators.

## 2.3 Crossing relations

Permuting the operators $S_{k_i}(x_i, t_i)$ in full correlator (2) imposes crossing relations between different channels. These consistency conditions relate CFT data appearing in possibly different OPEs (12). To recover familiar CFT crossing relations, we factorize the dynamical function as $\mathcal{G}_{k_1 k_2 k_3 k_4}(U, V; \sigma, \tau) = U^{k_{34}} \hat{\mathcal{G}}_{k_1 k_2 k_3 k_4}(U, V; \sigma, \tau)$. Swapping $1 \leftrightarrow 2$ gives the relation

$$\hat{\mathcal{G}}_{k_1 k_2 k_3 k_4}(U, V; \sigma, \tau) = \frac{1}{V^{k_3 - k_4}} \hat{\mathcal{G}}_{k_2 k_1 k_3 k_4}\left(\frac{U}{V}, \frac{1}{V}; \tau, \sigma\right), \tag{18}$$

while swapping $1 \leftrightarrow 3$ gives

$$\hat{\mathcal{G}}_{k_1 k_2 k_3 k_4}(U, V; \sigma, \tau) = \frac{U^{k_1 + k_2}}{V^{k_2 + k_3}} \tau^{\frac{k_1 + k_2 + k_3 - k_4}{2}} \hat{\mathcal{G}}_{k_3 k_2 k_1 k_4}\left(V, U; \frac{\sigma}{\tau}, \frac{1}{\tau}\right). \tag{19}$$

As usual, the condition derived by swapping $1 \leftrightarrow 2$ is trivially satisfied block-wise by the known form of 6d bosonic blocks [33, 34]. However, the $1 \leftrightarrow 3$ relation is not manifestly satisfied and is the fundamental constraint of the conformal bootstrap.

# 3 The 6d/$\mathcal{W}$-algebra conjecture

## 3.1 Solutions to the superconformal Ward identities

The second consequence of (8) is the observation that for certain configurations in cross-ratio space, the four-point function becomes topological along certain directions. By correlating space-time and R-symmetry cross-ratios with the twist $\alpha = \bar{\alpha} = \bar{z}^{-1}$, the superconformal Ward identity reduces to

$$\partial_{\bar{z}} \mathcal{G}_{k_1 k_2 k_3 k_4}\left(z, \bar{z}; \bar{z}^{-1}, \bar{z}^{-1}\right) = 0, \tag{20}$$

implying that this "twisted correlator" is a meromorphic function

$$\mathcal{G}_{k_1 k_2 k_3 k_4}\left(z, \bar{z}; \bar{z}^{-1}, \bar{z}^{-1}\right) \equiv \mathcal{F}_{k_1 k_2 k_3 k_4}(z).$$

This motivates the decomposition of full solutions to (8) into

$$\mathcal{G}_{k_1 k_2 k_3 k_4}(z, \bar{z}; \alpha, \bar{\alpha}) = \mathcal{F}_{k_1 k_2 k_3 k_4}(z, \bar{z}; \alpha, \bar{\alpha}) + \Upsilon \circ \mathcal{H}_{k_1 k_2 k_3 k_4}(z, \bar{z}; \alpha, \bar{\alpha}), \tag{21}$$

with inhomogeneous part $\mathcal{F}_{k_1 k_2 k_3 k_4}(z, \bar{z}; \alpha, \bar{\alpha})$ and a homogeneous part $\Upsilon \circ \mathcal{H}_{k_1 k_2 k_3 k_4}(z, \bar{z}; \alpha, \bar{\alpha})$ that vanishes identically upon twisting. The differential operator $\Upsilon$ is defined in [20] (with typos corrected in [31]) and the "reduced correlator" $\mathcal{H}_{k_1 k_2 k_3 k_4}(U, V; \sigma, \tau)$ is a polynomial in $\sigma$ and $\tau$ of degree 2 less than $\mathcal{G}_{k_1 k_2 k_3 k_4}(U, V; \sigma, \tau)$. This decomposition and the twisted correlator $\mathcal{F}_{k_1 k_2 k_3 k_4}(z)$ were first observed for 6d $\mathcal{N} = (2, 0)$ theories in [20] and have turned out to be the consequence of the cohomological procedure introduced in [1, 35] implemented at the level of four-point correlators. We will see in the following subsection that $\mathcal{F}_{k_1 k_2 k_3 k_4}(z)$ has the structure of a four-point function in a 2d meromorphic CFT.

## 3.2 The chiral algebra twist

The twist $\alpha = \bar{\alpha} = \bar{z}^{-1}$ can be applied to the full correlator (2) by first restricting operators to lie on a fixed $\mathbb{R}^2 \in \mathbb{R}^6$ with complex coordinates $z, \bar{z}$, i.e. taking $x_i = \left(\frac{1}{2}(z_i + \bar{z}_i), \frac{1}{2i}(z_i - \bar{z}_i), \mathbf{0}\right)$. The twist is then implemented by further correlating the R-symmetry polarizations with plane coordinates such that $t_{ij} = \bar{z}_{ij}^2$. As described in [1], this procedure is a reduction $\rho$ of the full 6d $\mathcal{N} = (2,0)$ operator algebra to the cohomology of a certain nilpotent supercharge $\mathbf{Q}$. Among the list of 6d operators we consider in (13), the only operators that survive this reduction are found in the supermultiplets

$$\mathcal{D}[p,0], \qquad \mathcal{D}[p,2], \qquad \mathcal{B}[p,0]_\ell. \qquad (22)$$

In particular, only a subset of the BPS $\mathcal{D}$ and semi-short $\mathcal{B}$ supermultiplets will play a role, while unprotected $\mathcal{L}$ supermultiplets do not survive. This restricts the selection rules in (12) to

$$\mathcal{D}[k_1,0] \otimes \mathcal{D}[k_2,0]\Big|_\rho = \bigoplus_{i=0}^{\min\{k_1,k_2\}} \mathcal{D}[k_1 + k_2 - 2i, 0]\Big|_\rho \oplus \bigoplus_{i=0}^{\min\{k_1,k_2\}} \mathcal{D}[k_1 + k_2 - 2i - 2, 2]\Big|_\rho$$
$$\oplus \bigoplus_{i=1}^{\min\{k_1,k_2\}} \bigoplus_{\ell=0}^{\infty} \mathcal{B}[k_1 + k_2 - 2i, 0]_\ell\Big|_\rho, \qquad (23)$$

where we additionally apply the physical selection rules described in Section 2.2 as appropriate. The reduction $\rho$ maps conformal multiplets in $\mathfrak{osp}(8^*|4)$ representations to global $\mathfrak{sl}(2,\mathbb{R})$ representations $\nu_{\Delta,\ell}$. For the operators we consider, the reduction maps [1]

$$\rho \; : \; \mathcal{D}[p,0] \mapsto \nu_{2p,0}, \qquad \mathcal{D}[p,2] \mapsto \nu_{2p+5,1}, \qquad \mathcal{B}[p,0]_\ell \mapsto \nu_{2p+6+\ell,\ell+2}. \qquad (24)$$

We can see this explicitly at the level of individual 6d $\mathcal{N} = (2,0)$ superblocks in (11). In this construction, only $\mathbf{Q}$-closed operators emerge as meromorphic operators in the anticipated chiral algebra. The twist was such that any dependence on the anti-holomorphic cross-ratio $\bar{\chi} = \frac{\bar{z}_{12}\bar{z}_{34}}{\bar{z}_{13}\bar{z}_{24}}$ is $\mathbf{Q}$-exact and therefore does not contribute to the cohomological reduction $\rho$. We can organize the $\bar{\chi}$-dependence of a twisted 6d superblock in a small $\bar{\chi}$ expansion.[5] We identify the $\bar{\chi}$-independent piece as the corresponding $SL(2,\mathbb{R})$ block in the chiral algebra.

Indeed, the superblocks associated with (22) have the small-$\bar{\chi}$ expansion

$$\lim_{\bar{\chi}\to 0} \mathfrak{G}_{\mathcal{D}[p,0]}^{\Delta_{12},\Delta_{34}}\left(\chi, \bar{\chi}; \bar{\chi}^{-1}, \bar{\chi}^{-1}\right) = \mathcal{Y}_{\frac{p}{2}\frac{p}{2}}^{k_{12},k_{34}} g_{2p,0}^{\Delta_{12},\Delta_{34}}(\chi) + O(\bar{\chi}),$$

$$\lim_{\bar{\chi}\to 0} \mathfrak{G}_{\mathcal{D}[p,2]}^{\Delta_{12},\Delta_{34}}\left(\chi, \bar{\chi}; \bar{\chi}^{-1}, \bar{\chi}^{-1}\right) = \mathcal{Y}_{\frac{p}{2}+1\frac{p}{2}+1}^{k_{12},k_{34}} C_{\mathcal{O}_{2p+5,1,\frac{p}{2}+1\frac{p}{2}+1}}^{\mathcal{D}[p,2],\Delta_{12},\Delta_{34}} g_{2p+5,1}^{\Delta_{12},\Delta_{34}}(\chi) + O(\bar{\chi}), \qquad (25)$$

$$\lim_{\bar{\chi}\to 0} \mathfrak{G}_{\mathcal{B}[p,0]_\ell}^{\Delta_{12},\Delta_{34}}\left(\chi, \bar{\chi}; \bar{\chi}^{-1}, \bar{\chi}^{-1}\right) = \mathcal{Y}_{\frac{p}{2}+1\frac{p}{2}+1}^{k_{12},k_{34}} C_{\mathcal{O}_{2p+6+\ell,\ell+2,\frac{p}{2}+1\frac{p}{2}+1}}^{\mathcal{B}[p,0]_\ell,\Delta_{12},\Delta_{34}} g_{2p+6+\ell,\ell+2}^{\Delta_{12},\Delta_{34}}(\chi) + O(\bar{\chi}).$$

We see that in this limit, the 6d blocks reduce to global $SL(2,\mathbb{R})$ blocks defined in Appendix A which represent exchanged conformal multiplets with a quasi-primary of holomorphic conformal weight $h = \frac{\Delta+\ell}{2}$. The factors $\mathcal{Y}_{mn}^{k_{12},k_{34}}$ are the leading coefficients in the small-$\bar{\chi}$ expansion of the $Y_{mn}^{a,b}(\sigma,\tau)$ polynomials:

$$\lim_{\bar{\chi}\to 0} Y_{\frac{p}{2}\frac{p}{2}}^{\frac{|k_{12}-k_{34}|}{2}, \frac{|k_{23}-k_{14}|}{2}}\left(\bar{\chi}^{-2}, \left(1 - \bar{\chi}^{-1}\right)^2\right) = \bar{\chi}^{-2p}\left(\mathcal{Y}_{\frac{p}{2}\frac{p}{2}}^{k_{12},k_{34}} + O\left(\bar{\chi}^{-1}\right)\right). \qquad (26)$$

---

[5]This follows the analysis of [36] which was applied to superblock decompositions of $\langle S_p S_p S_p S_p \rangle$.

The $\bar{\chi}$-dependence in $G^{\Delta_{12},\Delta_{34}}_{\Delta,\ell}$ then precisely cancel the negative powers of $\bar{\chi}$ to give the result (25). For the configurations appearing in this work, our conventions result in

$$\mathcal{Y}^{k_{12},k_{34}}_{\frac{p}{2}\frac{p}{2}} = \frac{p!\left(\frac{k_{12}-k_{34}}{2}\right)!}{\left(\frac{p+k_{12}}{2}\right)!\left(\frac{p-k_{34}}{2}\right)!}, \tag{27}$$

which coincide with the reciprocal of the $s$-channel R-symmetry gluing factors mentioned in Equation (3.13) of [17], as well as their crossed analogues. The coefficients $\mathcal{C}^{\mathcal{X},\Delta_{12},\Delta_{34}}_{\mathcal{O}_{\Delta,\ell,mn}}$ are computable on case-by-case basis, e.g. using the strategy in [21–23]. For example, we found by studying many cases that the only such coefficient appearing in $\langle S_p S_p S_q S_q \rangle$ obeys

$$\mathcal{C}^{\mathcal{B}[2n,0]_\ell,0,0}_{\mathcal{O}_{4n+6+\ell,\ell+2,n+1n+1}} = \frac{n+1}{2(2n+1)}\frac{\ell+3}{\ell+1}. \tag{28}$$

We also find that

$$\mathcal{C}^{\mathcal{D}[n,2],4-2n,4-2n}_{\mathcal{O}_{2n+5,1,\frac{n}{2}+1\frac{n}{2}+1}} = -\mathcal{C}^{\mathcal{D}[n,2],2n-4,4-2n}_{\mathcal{O}_{2n+5,1,\frac{n}{2}+1\frac{n}{2}+1}} = \frac{4n}{(n+1)(n+2)}, \tag{29}$$

$$\mathcal{C}^{\mathcal{B}[n,0]_\ell,4-2n,4-2n}_{\mathcal{O}_{2n+6+\ell,\ell+2,\frac{n}{2}+1\frac{n}{2}+1}} = (-1)^\ell \mathcal{C}^{\mathcal{B}[n,0]_\ell,2n-4,4-2n}_{\mathcal{O}_{2n+6+\ell,\ell+2,\frac{n}{2}+1\frac{n}{2}+1}} = \frac{2n}{(n+1)(n+2)}\frac{\ell+3}{\ell+1}, \tag{30}$$

which are useful e.g. for studying $\langle S_2 S_p S_2 S_p \rangle$. In $\langle S_3 S_3 S_2 S_4 \rangle$ and $\langle S_2 S_4 S_4 S_4 \rangle$ we need

$$\mathcal{C}^{\mathcal{B}[2,0]_\ell,-4,0}_{\mathcal{O}_{10+\ell,\ell+2,2\,2}} = \mathcal{C}^{\mathcal{B}[2,0]_\ell,0,-4}_{\mathcal{O}_{10+\ell,\ell+2,2\,2}} = \frac{1}{2}\frac{\ell+3}{\ell+1}, \tag{31}$$

$$\mathcal{C}^{\mathcal{B}[4,0]_\ell,-4,0}_{\mathcal{O}_{14+\ell,\ell+2,3\,3}} = \mathcal{C}^{\mathcal{B}[4,0]_\ell,0,-4}_{\mathcal{O}_{14+\ell,\ell+2,3\,3}} = \frac{2}{5}\frac{\ell+3}{\ell+1}, \tag{32}$$

which concludes the list of such coefficients used in this work.

So the chiral algebra twist devolves the 6d superblock expansion (10) into the $SL(2,\mathbb{R})$ block expansion

$$\mathcal{F}_{k_1 k_2 k_3 k_4}(\chi) = \delta_{k_1 k_2}\delta_{k_3 k_4} + \sum_p \lambda_{k_1 k_2 \mathcal{D}[p,0]}\lambda_{k_3 k_4 \mathcal{D}[p,0]}\mathcal{Y}^{k_{12},k_{34}}_{\frac{p}{2}\frac{p}{2}}g^{\Delta_{12},\Delta_{34}}_{2p,0}(\chi)$$

$$+ \sum_p \lambda_{k_1 k_2 \mathcal{D}[p,2]}\lambda_{k_3 k_4 \mathcal{D}[p,2]}\mathcal{Y}^{k_{12},k_{34}}_{\frac{p}{2}+1\frac{p}{2}+1}\mathcal{C}^{\mathcal{D}[p,2],\Delta_{12},\Delta_{34}}_{\mathcal{O}_{2p+5,1,\frac{p}{2}+1\frac{p}{2}+1}}g^{\Delta_{12},\Delta_{34}}_{2p+5,1}(\chi) \tag{33}$$

$$+ \sum_p \sum_\ell \lambda_{k_1 k_2 \mathcal{B}[p,0]_\ell}\lambda_{k_3 k_4 \mathcal{B}[p,0]_\ell}\mathcal{Y}^{k_{12},k_{34}}_{\frac{p}{2}+1\frac{p}{2}+1}\mathcal{C}^{\mathcal{B}[p,0]_\ell,\Delta_{12},\Delta_{34}}_{\mathcal{O}_{2p+6+\ell,\ell+2,\frac{p}{2}+1\frac{p}{2}+1}}g^{\Delta_{12},\Delta_{34}}_{2p+6+\ell,\ell+2}(\chi),$$

where the $p$ sums include all supermultiplets

$$\mathcal{X} \in \left(\mathcal{D}[k_1,0]\times\mathcal{D}[k_2,0]\right)\big|_\rho \cap \left(\mathcal{D}[k_3,0]\times\mathcal{D}[k_4,0]\right)\big|_\rho,$$

as allowed by (23). This is an invitation to interpret 6d data $\lambda_{k_1 k_2 \mathcal{X}}\lambda_{k_3 k_4 \mathcal{X}}$ in terms of an auxiliary (and inevitably more tractable) 2d meromorphic CFT with four-point function

$$\mathcal{F}_{k_1 k_2 k_3 k_4}(\chi) = \sum_{\mathcal{O},\mathcal{O}'} C^{\mathcal{O}}_{k_1 k_2} G_{\mathcal{O}\mathcal{O}'} C^{\mathcal{O}'}_{k_3 k_4} g^{\Delta_{12},\Delta_{34}}_{\Delta_{\mathcal{O}},\ell_{\mathcal{O}}}(\chi). \tag{34}$$

We briefly stop to point out an important coarsening that occurs when passing from (10) to (33) using the reduction (24). While 6d superconformal multiplets and their corresponding superblocks depend on two quantum numbers $(\Delta,\ell)$, the $\mathfrak{sl}(2,\mathbb{R})$ representations and the global

blocks in the reduction only depend on the combination $h = \frac{\Delta + \ell}{2}$.[6,7] This makes an explicit mapping between 6d operators and 2d quasi-primaries ambiguous and additional principles, such as crossing symmetry and the 6d OPE considerations in Section 2.2, are needed to correctly deduce 6d CFT data from this reduction. In Section 5.2 we will describe the practical obstacle that this poses when extracting 6d three-point coefficients $\lambda_{k_1 k_2 \mathcal{X}} \lambda_{k_3 k_4 \mathcal{X}}$, as well as the strategy we use to mostly overcome this.

In the following section, we will elaborate on the particular chiral algebras that [1] conjectured are associated to 6d (2,0) theories of type $\mathfrak{g}$. This will provide the theory-specific input needed to interpret the spectrum of exchanged quasi-primaries $\mathcal{O}$ and evaluate 6d CFT data $\lambda_{k_1 k_2 \mathcal{X}} \lambda_{k_3 k_4 \mathcal{X}}$.

# 4 $\mathcal{W}$-algebras of type $\mathfrak{g}$

Given the dearth of intuition for interacting 6d (2,0) theories, the authors of [1] needed to interpret their corresponding chiral algebras using minimal physical input. Their only postulate was the widely-believed folk theorem that the ring of 1/2-BPS operators in a (2,0) theory of type $\mathfrak{g}$ is freely-generated by a set of elements in one-to-one correspondence with the Casimir invariants of $\mathfrak{g}$. The $\mathfrak{so}(5)_R$ highest-weight states that form this 1/2-BPS ring are precisely those that yield the $v_{2p,0}$ quasi-primaries in Equations (24) and (25). It was argued that these quasi-primaries are the complete set of generators of the chiral algebra. These spectral considerations led [1] to conjecture that for a 6d $\mathcal{N} = (2,0)$ theory of type $\mathfrak{g}$, the meromorphic CFT obtained by the cohomological reduction $\rho$ is a quantum $\mathcal{W}_{\mathfrak{g}}$ algebra with central charge

$$c_{2d}(\mathfrak{g}) = c_{\mathfrak{g}} = 4 d_{\mathfrak{g}} h_{\mathfrak{g}}^{\vee} + r_{\mathfrak{g}}, \tag{35}$$

where the proportionality factor $c_{2d}(\mathfrak{g})/c_{\mathfrak{g}}$ is fixed by the abelian (2,0) theory and its reduction to a $\mathfrak{u}(1)$ affine current algebra. By $\mathcal{W}_{\mathfrak{g}}$, we mean a $\mathcal{W}$-algebra obtained via quantum Drinfel'd-Sokolov reduction of an affine Kac-Moody algebra $\hat{\mathfrak{g}}$.[8] Loosely speaking, this procedure takes $\hat{\mathfrak{g}}$ and imposes constraints on its generators with a BRST procedure. The OPEs of the fields in the resulting BRST cohomology close on the algebra $\mathcal{W}_{\mathfrak{g}}$.

For our purposes, it will be sufficient to view $\mathcal{W}_{\mathfrak{g}}$ as a higher-spin extension of the Virasoro algebra. In particular, we start with the weight-2 Virasoro generator $T$ and add $r_{\mathfrak{g}} - 1$ additional generators $W_{k_i}$ with conformal weights $\{k_i\}$ corresponding to degrees of the Casimir invariants of $\mathfrak{g}$. We will work in a basis where the additional generating currents $\{W_{k_i}\}$ are Virasoro primaries with respect to $T$.[9] Almost all quantum Drinfel'd Sokolov reductions $\mathcal{W}_{\mathfrak{g}}$ are known to be subalgebras of truncations of a larger algebra $\mathcal{W}_{1+\infty}[\lambda]$,[10] which additionally contains a spin-1 generator $W_1$ and depends on a central charge $c$ and a parameter $\lambda$ which are specified upon truncation, up to triality symmetry [47]. In this sense, $\mathcal{W}_{1+\infty}[\lambda]$ interpolates between

---

[6]The fundamental issue, called the filtration problem, is that while 6d **Q**-exact operators are labeled by $\mathfrak{so}(6)$ Lorentz quantum numbers $[c_1 = 0, c_2, c_3]$, $\mathfrak{so}(5)_R$ Dynkin labels $[d_1, d_2 = 0]$, and a scaling dimension fixed by $\Delta = 2d_1 + c_2 + \frac{c_3}{2}$ [1], the twisted translations used to define the cohomology violate the Cartan of an $\mathfrak{su}(2)$ subalgebra of $\mathfrak{so}(5)_R$, meaning that the reduction $\rho$ only preserves the quantum number $h = \frac{2d_1 + c_2 + \frac{c_3}{2}}{2}$. The 6d operators appearing in this work obey $c_3 = 0$ and we write $c_2 = \ell$.

[7]The analogous ambiguity occurs in the 4d $\mathcal{N} = 2$ chiral algebra construction of [35]. It was observed in [37] that $\mathfrak{su}(2)_R$ is violated in a sign-definite way, providing an organizational principle called $R$ filtration. Proposed derivations of this filtration and the related Macdonald refinement of the Schur limit of the 4d $\mathcal{N} = 2$ superconformal index from a purely VOA perspective have been made in e.g. [38–43].

[8]There are other ways to arrive at $\mathcal{W}_{\mathfrak{g}}$, e.g. the coset construction, as described pedagogically in [44, 45].

[9]There are other bases for $\mathcal{W}$-algebra generators such as the "quadratic basis", which bounds the non-linearity of the operator algebra to be quadratic. While less natural from a 6d perspective, the quadratic basis proposes closed-form expressions for generic OPE coefficients, e.g. for $\mathfrak{g} = A_{N-1}$ in [46].

[10]This is not known for $F_4$, which does not affect the study of the ADE-classified 6d (2,0) theories.

every $\mathcal{W}_{\mathfrak{g}}$ algebra relevant to our study and we can always obtain $\mathcal{W}_{\mathfrak{g}}$ by a certain truncation of $\mathcal{W}_{1+\infty}[\lambda]$. This allows us to follow [46, 47] and use triality and universality properties of $\mathcal{W}_{1+\infty}[\lambda]$ to parameterize candidate 6d (2,0) $\mathcal{W}_{\mathfrak{g}}$ algebras.

$\mathcal{W}_{\mathfrak{g}}$ symmetry has appeared in other 6d $\mathcal{N} = (2,0)$ theoretical contexts, the most famous of which being the AGT correspondence [48–50], where a 6d (2,0) theory of type $\mathfrak{g}$ is compactified on a complex curve $\mathcal{C}$ and its remaining four non-compact directions are subjected to an $\Omega_{\epsilon_1,\epsilon_2}$ deformation. The correspondence relates the partition function of the resulting theory to correlators in a Toda theory exhibiting $\mathcal{W}_{\mathfrak{g}}$ symmetry and a central charge that matches (35) when $\epsilon_1 = \epsilon_2$. Further evidence that the $\mathcal{W}$-algebra conjecture of [1] is compatible with this structure includes [51], which derives the aforementioned Toda theory via a dimensional reduction of the (2,0) theory on $S^4$ as well as [52], which provides an alternative derivation of the $\mathcal{W}_{\mathfrak{g}}$ algebra discussed in this work using an $\Omega_{\epsilon_1,\epsilon_2}$ deformation of the holomorphic topological twist of the (2,0) theory.

In the following subsections, we will briefly review the VOA bootstrap, before elucidating the $\mathcal{W}$-algebras conjecturally related to (2,0) theories of type $\mathfrak{g}$.[11] In Appendix B, we list the explicit $\mathcal{W}_{A_{N-1}}$ and $\mathcal{W}_{D_N}$ structure constant relations appearing in the correlators studied in this work.[12] In addition, we present associativity relations for some low-lying structure constants in the $\mathcal{W}$-algebras of exceptional type $E_6, E_7, E_8$.

## 4.1 VOA bootstrap generalities

Given a list of generating fields $\{W_i\}$, one proposes an operator algebra by writing down ansätze for the OPEs between each generator, i.e. the set of Laurent series

$$W_i(z)W_j(0) \sim \sum_{\alpha=-\infty}^{h_i+h_j} \frac{\{W_iW_j\}_\alpha(z)}{z^\alpha}\,. \tag{36}$$

Such ansätze are completely determined by the structure constants $C_{ij}^k$ appearing in the singular parts $\{W_iW_j\}_{\alpha>0}$ of this expression. The first regular term $\{W_iW_j\}_0$ is the normal-ordered product of $W_i$ and $W_j$ and one obtains quasi-primary and Virasoro primary projections of any term $\{W_iW_j\}_\alpha$ by taking appropriate subtractions. Following the conventions of [46, 47], we refer to the Virasoro primary involving the normal-ordered product $: \mathcal{O}_1\mathcal{O}_2\cdots\mathcal{O}_m :$ with $n$ derivatives as $[\mathcal{O}_1\mathcal{O}_2\cdots\mathcal{O}_m]^{(n)}$. We refer the reader to [46] and Appendix A of [38] for further review of VOA OPEs and useful formulae.

The crucial point is that not just any set of structure constants $\{C_{ij}^k\}$ will yield a consistent chiral algebra. The structure constants are constrained by the associativity of the VOA which in terms of the OPE (36), is set of Jacobi identities

$$\left\{W_i\left\{W_jW_k\right\}_\alpha\right\}_\beta - \left\{W_j\left\{W_iW_k\right\}_\beta\right\}_\alpha = \sum_{\gamma>0}\binom{\beta-1}{\gamma-1}\left\{\left\{W_iW_j\right\}_\gamma W_k\right\}_{\alpha+\beta-\gamma}\,, \tag{37}$$

which we will collectively refer to as $\left(W_iW_jW_k\right)$. We can solve associativity constraints and carry out various other OPE calculations seamlessly using the `Mathematica` package `OPEdefs` and its extension `OPEconf` introduced in [55, 56]. While the $SL(2,\mathbb{R})$ global blocks in (33) suggest that twisted 6d operators are most naturally thought of in terms of quasi-primaries, it is most efficient to express OPE ansätze (36) in terms of Virasoro primaries. The `MakeOPEP` command automatically computes coefficients of all normal-ordered products involving $T$ as

---

[11]Shortly after this work was released, two papers [53, 54] appeared, which analyzed the super $\mathcal{W}$-algebras that arise in the chiral algebra reduction of 4d $\mathcal{N} = 4$ SCFTs using similar VOA bootstrap techniques.

[12]These relations match those appearing in [46, 47], however our results for $\mathcal{W}_N$ are without the normalization or orthogonality conventions of [46], some of which are awkward from a 6d perspective.

determined by Virasoro symmetry. `OPEQPPole` can then be used to extract quasi-primary contributions to a given $\{W_i W_j\}_\alpha$.

Our 6d normalization conventions mean that the meromorphic stress tensor obtained by twisting $S_2$ is rescaled compared to the canonically normalized Virasoro generator $T$ encoded in `OPEdefs`, i.e.

$$S_2(x,t)\big|_\rho \equiv \hat{T}(\chi) = \sqrt{\frac{2}{c}}\, T(\chi). \tag{38}$$

Our 6d conventions also imply that generators $W_{k_i}$ are unit-normalized, i.e. $C^0_{ij} = \delta_{ij}$. However, the VOA bootstrap results presented in Appendix B and the ancillary `Mathematica` file will keep field normalizations and shift freedom unfixed for reference.

Finally, we note that the weight-1 current in $\mathcal{W}_{1+N}$ can always be factored out and we will therefore consider neither $W_1$ nor normal-ordered products thereof in our OPE ansätze. This is the VOA version of the statement that the 6d superconformal primary $S_1$ of $\mathcal{D}[1,0]$ decouples from the spectrum of operators in interacting (2,0) theories. Put differently, the map $\rho$ in (24) shows that any $\mathcal{D}[1,0]$ operator is necessarily captured by the chiral algebra and its redundancy in the $\mathcal{W}$-algebra is consistent with its exclusion from the 6d selection rules in Section 2.2.

## 4.2 $\mathcal{W}_{A_{N-1}} = \mathcal{W}_N$

The most widely studied family of $\mathcal{W}_{\mathfrak{g}}$ algebras are the $\mathcal{W}_N$ algebras associated with $\mathfrak{g} = A_{N-1}$. These are generated by the currents

$$\{T, W_3, W_4, \ldots, W_N\}.$$

With this spectrum, we write ansätze for all of the OPEs $W_i \times W_j$ for $i + j \leq 10$, which are summarized in Equations (39) and (46) of [46]. We use `OPEconf` to solve the associativity conditions $(W_i W_j W_k)$ for $i + j + k \leq 12$ to find the relations presented in Appendix B.1 and the ancillary `Mathematica` file. Upon rescaling generators and their normal-ordered composites to match our 6d conventions, we notice that each relation depends only on two parameters: the central charge $c$ and the squared structure constant $\left(C^4_{33}\right)^2$. In fact, it was proven in [57] that all higher OPEs $W_i \times W_j$ continue to be determined by these parameters and that the family of $\mathcal{W}_\infty$ algebras (of which $\mathcal{W}_N$ are quotients) can be parametrized by $c$ and $\left(C^4_{33}\right)^2$.

The authors of [58] analyzed the minimal representations of $\mathcal{W}_N$ to fix the $c$-dependence of the parameter $\left(C^4_{33}\right)^2$ for any $\mathcal{W}_N$ truncation of $\mathcal{W}_\infty[\lambda = N]$. The resulting relation between $N$, $c$, and $\left(C^4_{33}\right)^2$ is cubic in $N$, revealing a triality symmetry under which 3 different values of $N$ at fixed $c$ can give rise to the same algebra. This motivates the parameterization of $\mathcal{W}_N$ algebras introduced in [46], in terms of three roots $\lambda_i$:

$$\begin{aligned}
0 &= \lambda_1\lambda_2 + \lambda_1\lambda_3 + \lambda_2\lambda_3\,, \\
c &= (\lambda_1 - 1)(\lambda_2 - 1)(\lambda_3 - 1)\,, \\
\frac{\left(C^4_{33}\right)^2 C^0_{44}}{\left(C^0_{33}\right)^2} &= \frac{144}{c(5c+22)} \frac{(c+2)(\lambda_1 - 3)(\lambda_2 - 3)(\lambda_3 - 3)}{(\lambda_1 - 2)(\lambda_2 - 2)(\lambda_3 - 2)}\,.
\end{aligned} \tag{39}$$

We realize the central charge relation (35) by choosing $(\lambda_1, \lambda_2, \lambda_3) = (-2N, -2N, N)$, i.e.[13]

$$c_{2d}(A_{N-1}) = 4N^3 - 3N - 1 = c_{A_{N-1}}. \tag{40}$$

---

[13]The fixed $N$-dependence of $c$ in 6d (2,0) theories reduces the triality symmetry of $\mathcal{W}_N$ to the duality $\mathcal{W}_N[c] \simeq \mathcal{W}_{-2N}[c]$ for fixed $c$.

In our 6d normalization conventions, we then have

$$\left(C_{33}^4\right)^2 = \frac{144}{c(5c+22)}\frac{(c+2)(N-3)(c(N+3)+2(4N+3)(N-1))}{(N-2)(c(N+2)+(3N+2)(N-1))}\,. \tag{41}$$

It will be useful to express certain correlators in terms of $C_{44}^4$ for comparison with $D_N$ theory results, so from Appendix B.1 we quote that

$$C_{44}^4(A_{N-1}) = \frac{3(c+3)C_{33}^4}{(c+2)} - \frac{288(c+10)}{c(5c+22)C_{33}^4}\,. \tag{42}$$

### 4.3 $\mathcal{W}_{D_N}$

The $\mathcal{W}_{D_N}$ algebras should be generated by currents with weights labeled by the orders of Casimir invariants of $D_N$, so we write

$$\{T, W_4, W_6, \ldots, W_{2N-2}, p_N\}\,.$$

We can again encode this spectrum and the OPE ansätze into OPEconf and solve the associativity conditions. We will first ignore the $p_N$ generator and only consider the sector involving generators $\{T, W_i\}$, which only have even spin. The relevant structure constant relations are listed in Appendix B.2. It will be useful to view the $\{T, W_i\}$ sector as a truncation of the even-spin $\mathcal{W}_\infty^e$ algebras that arise e.g. in even spin minimal holography [59]. Upon fixing rescaling and shift symmetries we again find that all structure constant relations can be expressed in terms of $c$ and a parameter we can choose to be $(C_{44}^4)^2$. Analogously to $\mathcal{W}_N$, it was proven in [60] that $\mathcal{W}_\infty^e$ algebras exist in a two-parameter family, and so the operator algebra of the $\{T, W_i\}$ sector will depend only on $c$ and $(C_{44}^4)^2$ for any $N$.

To compute the $c$-dependence of $(C_{44}^4)^2$, the authors of [59] studied the three minimal representations $\phi_a$ of $\mathcal{W}_\infty^e$, each with weight $h_a$ and OPE ansätze

$$\begin{aligned}
W_4 \times \phi_a &= C_{4a}^a \phi_a\,,\\
W_6 \times \phi_a &= C_{6a}^a \phi_a + C_{6a}^{[4a]^{(0)}}[W_4\phi_a]^{(0)} + C_{6a}^{[4a]^{(1)}}[W_4\phi_a]^{(1)}\,.
\end{aligned} \tag{43}$$

The $\left(W_4 W_4 \phi_a\right)$ Jacobi identities give that

$$C_{4a}^a = \frac{-\left(h_a\left(ch_a+c+3h_a^2-7h_a+2\right)\left(2ch_a+c+16h_a^2-10h_a\right)\right)}{12\left(2c^2h_a^2-2c^2h_a-9c^2+36ch_a^3-147ch_a^2+120ch_a-6c+24h_a^3+10h_a^2-28h_a\right)}C_{44}^4\,, \tag{44}$$

while the $\left(W_4 W_6 \phi_a\right)$ Jacobi identities fix the form of $(C_{44}^4)^2/C_{44}^0$ in terms of $c$ and $h_a$.

This minimal representation analysis holds for any truncation of $\mathcal{W}_\infty^e$, such as $\mathcal{W}_{B_N}$ or $\mathcal{W}_{C_N}$. We specialize our discussion to $\mathcal{W}_{D_N}$ by identifying the minimal representation $\phi_3$ with the generator $p_N$, causing $p_N$ to inherit the OPE data in (43) and (44) [47, 59]. Although the generator $p_N$ of (potentially odd) weight $h_3 = N$ excludes $\mathcal{W}_{D_N}$ from being a truncation of $\mathcal{W}_\infty^e$, it can be embedded in $\mathcal{W}_{1+\infty}$. This means that $\mathcal{W}_{D_N}$ inherits the same triality symmetry as $\mathcal{W}_N$, motivating the parameterization introduced in [47] which expresses the central charge in terms of roots $\lambda_i$ as

$$c = -\frac{(\lambda_1 + 2\lambda_2(\lambda_1+1))(\lambda_2+2\lambda_1(\lambda_2+1))}{\lambda_1+\lambda_2}\,. \tag{45}$$

Taking the same choice of roots $(\lambda_1, \lambda_2, \lambda_3) = (-2N, -2N, N)$, we find

$$c_{2d}(D_N) = 16N^3 - 24N^2 + 9N = c_{D_N}\,, \tag{46}$$

which is exactly the conjectured relationship (35) for $\mathfrak{g} = D_N$.

In our 6d normalizations, $\left(W_4 W_6 \phi_a\right) \longleftrightarrow \left(W_4 W_6 p_N\right)$ then gives us that

$$C_{44}^4 (D_N)^2 = \frac{144}{c(5c+22)} \tag{47}$$
$$\times \frac{\left(c^2(2(N-1)N-9)+3c(N(N(12N-49)+40)-2)+2N(N(12N+5)-14)\right)^2}{((c-7)N+c+3N^2+2)(c(N-2)(2N-3)+N(4N-5))(2cN+c+2N(8N-5))}.$$

We point out that the $p_N$ generator is expected from a 6d perspective, in that it is the twisted version of the Pfaffian superconformal primary $P_N$ of the $\mathcal{D}[N,0]$ supermultiplet that is expected in the SCFT dual to the M-theory orbifold $AdS_7 \times S^4/\mathbb{Z}_2$ [6, 7].

### 4.4 Accidental isomorphisms

Isomorphisms between the $A_{N-1}$ and $D_N$ algebras for low $N$ imply that their associated 6d (2,0) theories should coincide. As a basic consistency check, we can see this working at the level of the $\mathcal{W}$-algebras constructed above.

**$D_1 \simeq \mathfrak{u}(1)$:** The algebra $\mathcal{W}_{D_1}$ is generated only by the Pfaffian operator $p_1$ of weight 1, making it isomorphic to the $\mathfrak{u}(1)$ affine current algebra obtained by twisting the abelian (2,0) theory.

**$D_2 \simeq A_1 \times A_1$:** Here, the central charge relation (35) gives

$$c_{D_2} = 50 = c_{A_1,1} + c_{A_1,2}. \tag{48}$$

In the primary basis, the $\mathcal{W}_{D_2}$ algebra is generated by a Virasoro generator $T_{D_2}$ and Pfaffian primary $p_2$. On the other hand, $\mathcal{W}_{A_1 \times A_1}$ is generated by two decoupled Virasoro generators $T_{A_1,1}$ and $T_{A_1,2}$. We reconcile these pictures by identifying $T_{D_2} = T_{A_1,1} + T_{A_1,2}$ and $p_2 = T_{A_1,1} - T_{A_1,2}$. A short OPE calculation then shows that $p_2$ is a primary with respect to $T_{D_2}$, thanks to the condition that $c_{A_1,1} = c_{A_1,2}$.

**$D_3 \simeq A_3$:** In this case, the Pfaffian operator $p_3$ is isomorphic to the $W_3$ current of $\mathcal{W}_4$ while $W_4$ plays the same role in both algebras. Indeed, using the central charge relation (35)

$$c_{D_3} = 243 = c_{A_3}, \tag{49}$$

while the OPE data (41), (44), (B.1), (42), and (47) give

$$C_{4p_3}^{p_3}(D_3)^2 = \frac{11858}{166995} = C_{34}^3(A_3)^2,$$
$$C_{44}^4(D_3)^2 = \frac{2341448}{204568875} = C_{44}^4(A_3)^2. \tag{50}$$

Incidentally, the correlators studied in this work exhaust every possible four-point function of strong generators in these low-rank theories. The explicit form of the twisted correlators $\mathcal{F}_{k_1 k_2 k_3 k_4}$ in the ancilliary `Mathematica` file demonstrates the equivalence of each pair of isomorphic twisted theories.

## 5 6d (2,0) CFT data from the holomorphic bootstrap

Having identified a twisted subsector of 6d (2,0) operators with a $\mathcal{W}_{\mathfrak{g}}$ algebra, we can study twisted 6d correlators $\mathcal{F}_{k_1 k_2 k_3 k_4}(\chi)$ using the power of meromorphic CFT. In particular, we wish to compute the OPE coefficients $\lambda_{k_1 k_2 \mathcal{X}} \lambda_{k_3 k_4 \mathcal{X}}$ for the supermultiplets

$\mathcal{X} \in \{\mathcal{D}[p,0], \mathcal{D}[p,2], \mathcal{B}[p,0]_\ell\}$ that survive the reduction $\rho$. We will do this by comparing the twisted block decomposition (33) with a closed-form expression for $\mathcal{F}_{k_1 k_2 k_3 k_4}(\chi)$ obtained using the holomorphic bootstrap method of [2].

## 5.1 Holomorphic bootstrap

Following the "holomorphic bootstrap" method of [2], we can use crossing symmetry and the observation that the only singularities of $\mathcal{F}_{k_1 k_2 k_3 k_4}(\chi)$ in the complex $\chi$-plane are at $\chi = 0, 1, \infty$, to obtain closed-form expressions for $\mathcal{F}_{k_1 k_2 k_3 k_4}(\chi)$. In particular, for four-point functions of chiral quasi-primaries of weight $h_i = k_i$, we have that

$$\mathcal{F}_{k_1 k_2 k_3 k_4}(\chi) = G_{k_1 k_2} G_{k_3 k_4} + \sum_{n=1}^{k_1+k_4} \alpha_{n,\{k_i\}} \chi^n + \sum_{n=1}^{k_2+k_3} \beta_{n,\{k_i\}} \left( (1-\chi)^{-n} - 1 \right). \tag{51}$$

The coefficients $\alpha_{n,\{k_i\}}$ and $\beta_{n,\{k_i\}}$ are determined by the singular behavior at $\chi = \infty$ and $\chi = 1$, which are accessed by crossing relations to the $1 \leftrightarrow 4$ and $2 \leftrightarrow 4$ permutations, respectively. Explicitly, we have

$$\alpha_{n,\{k_i\}} = (-1)^\Sigma \sum_{h=0}^{k_1+k_4-n} \sum_{\mathcal{O}(h),\mathcal{O}'(h)} C^{\mathcal{O}}_{k_4 k_2} C^{\mathcal{O}'}_{k_3 k_1} G_{\mathcal{O}\mathcal{O}'} \frac{(h-k_{42})_{k_1+k_4-h-n}(h-k_{31})_{k_1+k_4-h-n}}{(k_1+k_4-h-n)!\,(2h)_{k_1+k_4-h-n}}, \tag{52}$$

$$\beta_{n,\{k_i\}} = (-1)^\Sigma \sum_{h=0}^{k_2+k_3-n} \sum_{\mathcal{O}(h),\mathcal{O}'(h)} C^{\mathcal{O}}_{k_1 k_4} C^{\mathcal{O}'}_{k_3 k_2} G_{\mathcal{O}\mathcal{O}'}$$
$$\times \sum_{l=0}^{k_2+k_3-h-n} \frac{(h-k_{14})_l (h-k_{23})_l (-k_3-k_4)_{k_2+k_3-h-n-l}}{(k_2+k_3-h-n-l)!\,(l)!\,(2h)_l}, \tag{53}$$

where $\Sigma = k_1 + k_2 + k_3 + k_4$. It is understood in these expressions that we sum over all possible pairs of exchanged quasi-primaries $\mathcal{O}, \mathcal{O}'$ appearing in the two OPEs, where both quasi-primaries have equal weight $h$, as in the $SL(2,\mathbb{R})$ block decomposition of a chiral algebra four-point function in (34). It will be impractical to diagonalize our basis of quasi-primaries and $G_{\mathcal{O}\mathcal{O}'}$ capture possibly off-diagonal two-point function coefficients. While not especially illuminating, the formulae for $\alpha_{n,\{k_i\}}, \beta_{n,\{k_i\}}$ demonstrate that meromorphicity and crossing symmetry constrain $\mathcal{F}_{k_1 k_2 k_3 k_4}(\chi)$ to depend only on a finite number of low-lying OPE coefficient products $C^{\mathcal{O}}_{ij} G_{\mathcal{O}\mathcal{O}'} C^{\mathcal{O}'}_{kl}$ appearing in the block decomposition (34). Our summary has thus far applied to a generic meromorphic CFT and theory-dependent input is needed to specify the spectrum of exchanged quasi-primaries and the OPE coefficients. We will henceforth specialize to the case of $\mathcal{W}_{\mathfrak{g}}$, where we normalize strong generators $W_i, W_j$ such that $G_{ij} = \delta_{ij}$.

Although crossing symmetry was used to write (51), the expression (51) does not manifestly solve the crossing equations. Upon twisting, the 6d crossing relations in Section 2.3 reduce to

$$\mathcal{F}_{k_1 k_2 k_3 k_4}(\chi) = \frac{1}{(1-\chi)^{k_3-k_4}} \mathcal{F}_{k_2 k_1 k_3 k_4}\left( \frac{\chi}{\chi-1} \right), \tag{54}$$

$$= \frac{\chi^{k_1+k_2}}{(1-\chi)^{k_2+k_3}} \mathcal{F}_{k_3 k_2 k_1 k_4}(1-\chi). \tag{55}$$

Demanding that $\mathcal{F}_{k_1 k_2 k_3 k_4}(\chi)$ satisfies (54) and (55) imposes non-trivial constraints on the structure constants $C^{\mathcal{O}}_{ij}$ appearing in $\alpha_{n,\{k_i\}}, \beta_{n,\{k_i\}}$ for $\{k_i\}$ and its $1 \leftrightarrow 4$ and $2 \leftrightarrow 4$ crossed configurations. These constraints will turn out to be the same as those derived using the VOA bootstrap. As a consistency check for our methodology, we conducted this crossing analysis for

all of the correlators in this work and obtained the same constraints on the $\mathcal{W}$-algebra structure constants as those computed using OPEconf and recorded in Appendix B. The upshot is that every twisted correlator can be expressed in terms of only $c(\mathfrak{g})$ and either $C_{33}^4$ or $C_{44}^4(\mathfrak{g})$. For example, we have

$$\mathcal{F}_{22pp}(\chi) = 1 + \delta_{2p}\left(\frac{1}{(1-\chi)^4} + 1\right)\chi^4 + \frac{2p\chi^2(\chi(p\chi - 2) + 2)}{c(\chi - 1)^2},$$

$$\mathcal{F}_{2p2p}(\chi) = \delta_{2p}\frac{\chi^{p-2}(2(\chi - 1)\chi((\chi - 1)\chi + 2) + 1)}{(\chi - 1)^4} + \frac{\chi^p\left(c(\chi - 1)^2\chi^2 + 2p^2 + 4p(\chi - 1)\chi\right)}{c(\chi - 1)^2},$$

$$\mathcal{F}_{2334}(\chi) = \frac{\sqrt{2}\chi^3(\chi(3\chi - 2) + 3)}{(\chi - 1)^2}\frac{C_{33}^4}{\sqrt{c}}, \tag{56}$$

$$\mathcal{F}_{2444}(\chi) = \frac{4\sqrt{2}\chi^4((\chi - 1)\chi + 1)}{(\chi - 1)^4}\frac{C_{44}^4}{\sqrt{c}},$$

$$\mathcal{F}_{3344}(\chi) = 1 - \frac{12(\chi(5\chi + 2) - 2)\chi^2}{c(\chi - 1)^2},$$

$$+ \frac{\chi^4\left(3(c + 2)\chi^4 - 8(c + 2)\chi^3 + (17c + 43)\chi^2 - 18(c + 3)\chi + 9(c + 3)\right)\left(C_{33}^4\right)^2}{3(\chi - 1)^4}\frac{}{c + 2}.$$

We have computed all $\mathcal{F}_{k_1k_2k_3k_4}(\chi)$ up to and including $\{k_i\} = 4$ and display them explicitly in the ancilliary Mathematica file. We point out that the authors of [18] use a similar bootstrap procedure to write down a closed-form expression for $\mathcal{F}_{k_1k_2k_3k_4}(\chi)$ to leading order in $c^{-1}$. The authors of [31] previously wrote down formulae for $\mathcal{F}_{kkkk}(\chi)$ up to $k = 4$ to leading order in $c^{-1}$, which match our results. An immediate consequence is that we can uplift the 2d correlator $\mathcal{F}_{k_1k_2k_3k_4}(\chi)$ to the 6d inhomogeneous correlator $\hat{\mathcal{F}}_{k_1k_2k_3k_4}(U, V; \sigma, \tau)$. The latter is constrained to be a polynomial of degree $\{k_i\}$ in $\sigma, \tau$ (times $\tau^\delta$), to satisfy the crossing relations (18) and (19), and to reduce to $\mathcal{F}_{k_1k_2k_3k_4}(\chi)$ upon twisting. We can apply these constraints to the examples in (56) to find[14]

$$\hat{\mathcal{F}}_{22pp}(U, V; \sigma, \tau) = 1 + \delta_{2p}\left(\sigma^2 U^4 + \frac{\tau^2 U^4}{V^4}\right) + \frac{2p}{c}\left(\frac{(p-1)\sigma\tau U^4}{V^2} + \sigma U^2 + \frac{\tau U^2}{V^2}\right),$$

$$\hat{\mathcal{F}}_{2p2p}(U, V; \sigma, \tau) = U^{p-2}\left(\sigma^2 U^4 + \delta_{2p}\left(1 + \frac{\tau^2 U^4}{V^4}\right) + \frac{2p}{c}\left(\frac{(p-1)\tau U^2}{V^2} + \frac{\sigma\tau U^4}{V^2} + \sigma U^2\right)\right),$$

$$\hat{\mathcal{F}}_{2334}(U, V; \sigma, \tau) = \frac{C_{33}^4 U^1}{\sqrt{c}}\left(\frac{2\sqrt{2}\sigma\tau U^4}{V^2} + \sqrt{2}\sigma U^2 + \frac{2\sqrt{2}\tau U^2}{V^2}\right),$$

$$\hat{\mathcal{F}}_{2444}(U, V; \sigma, \tau) = \frac{C_{44}^4}{\sqrt{c}}\left(\frac{2\sqrt{2}\sigma\tau U^4}{V^2} + \frac{2\sqrt{2}\tau^2 U^4}{V^4} + \frac{2\sqrt{2}\sigma\tau^2 U^6}{V^4}\right),$$

$$\hat{\mathcal{F}}_{3344}(U, V; \sigma, \tau) = 1 + \frac{12}{c}\left(\sigma U^2 + \frac{\tau U^2}{V^2} - 6\frac{\sigma\tau U^4}{V^2}\right)$$

$$+ \frac{\left(C_{33}^4\right)^2}{3}\left(\sigma^2 U^4 + \frac{\tau^2 U^4}{V^4} + \frac{2\sigma\tau^2 U^6}{V^4} + \frac{2\sigma^2\tau U^6}{V^2}\right)$$

$$+ \frac{(7c + 23)\left(C_{33}^4\right)^2}{3(c + 2)}\frac{\sigma\tau U^4}{V^2}, \tag{57}$$

where the hats are a reminder to reinstate the factor $U^{k_{34}}$ to obtain the $\mathcal{F}_{k_1k_2k_3k_4}(U, V; \sigma, \tau)$ contribution to $\mathcal{G}_{k_1k_2k_3k_4}(U, V; \sigma, \tau)$. The uplift and decomposition in (21) are not unique

---

[14]Note that our expression for $\mathcal{F}_{22pp}(U, V; \sigma, \tau)$ corrects a typo in Equation (2.17) of [23].

and one can always "improve" $\mathcal{F}_{k_1 k_2 k_3 k_4}(U, V; \sigma, \tau)$, e.g. to organize $\mathcal{G}_{k_1 k_2 k_3 k_4}(U, V; \sigma, \tau)$ in a reduced superblock expansion as done for $\langle S_2 S_2 S_p S_p \rangle$ in [23].

Before we proceed, it is instructive to contrast the relative utility of the chiral algebra and protected correlator $\mathcal{F}_{k_1 k_2 k_3 k_4}$ in 6d (2,0) theories with their analogues in 4d $\mathcal{N} = 4$ super Yang-Mills SCFT, where the chiral algebra is a certain non-unitary super $\mathcal{W}$-algebra [35]. Four-point correlators of 1/2-BPS superprimaries in 4d $\mathcal{N} = 4$ theories obey analogous superconformal Ward identities and can be decomposed as in (21), where the operator $\Upsilon^{(4d)}$ becomes a multiplicative factor and $\mathcal{F}^{(4d)}_{k_1 k_2 k_3 k_4}$ again encodes the holomorphic remainder after the twist. While $\mathcal{F}^{(4d)}_{k_1 k_2 k_3 k_4}$ can be determined using the underlying chiral algebra, the non-renormalization theorems of [61] dictate that one may just as well determine this contribution using Wick contractions in the free theory where $g_{\text{YM}} = 0$.[15] Constraints from the 4d $\mathcal{N} = 4$ chiral algebra twist become indispensable, however, when considering higher-point functions [64, 65], as well as four-point correlators involving external 1/4-BPS superprimaries [66].

Unlike 4d $\mathcal{N} = 4$ super Yang-Mills theory, 6d $\mathcal{N} = (2, 0)$ theories lack a coupling parameter or a conventional free field limit with which one can fix $\mathcal{F}_{k_1 k_2 k_3 k_4}$ and so the $\mathcal{W}$-algebra conjecture provides the only known principle for determining this contribution. From a holographic viewpoint, the 4d $\mathcal{N} = 4$ chiral algebra is agnostic to the 't Hooft coupling $\lambda^2 = g_{\text{YM}}^2 N$ and therefore does not encode stringy corrections to supergravity in the type IIB dual on $AdS_5 \times S^5$. By contrast, 6d $\mathcal{N} = (2, 0)$ $\mathcal{W}$-algebra data has an infinite expansion in non-trivial powers of $c_{6d}^{-1}$, which we will interpret holographically in Section 6 as describing protected M-theory higher-derivative corrections to supergravity on $AdS_7 \times S^4 / \mathbb{Z}_{\mathfrak{o}}$.

## 5.2 Extracting 6d CFT data

Equipped with a closed-form expression for $\mathcal{F}_{k_1 k_2 k_3 k_4}(\chi)$ in (51), we can simply compare this with the twisted 6d block expansion (33) in a small-$\chi$ expansion to obtain 6d data $\lambda_{k_1 k_2 \mathcal{X}} \lambda_{k_3 k_4 \mathcal{X}}$ in terms of minimal $\mathcal{W}_{\mathfrak{g}}$ data.

An immediate issue is the filtration ambiguity explained at the end of Section 3.2. At the level of determining $\lambda_{k_1 k_2 \mathcal{X}} \lambda_{k_3 k_4 \mathcal{X}}$, the basic problem is that there are as many, if not more ways of building quasi-primaries of fixed $h = \frac{\Delta + \ell}{2}$ from $W_i$, as there are 6d twisted superconformal multiplets allowed by the selection rules (23). Moreover, 6d supermultiplets that would normally be distinguishable by a twist parameter $\bar{h} = \frac{\Delta - \ell}{2}$ become degenerate when only parameterized by $h$. For example, the 6d superprimary $\mathcal{D}[6, 0]$ and the $\mathbf{Q}$-exact superdescendant $(\Delta, \ell) = (10, 2)$ of $\mathcal{B}[2, 0]_0$ both have $h = 6$ and can a priori correspond to any of the 2d operators $W_6, [T T]^{(2)}, [T W_3]^{(1)}, [T W_4]^{(0)}, [W_3 W_3]^{(0)}, [T T T]^{(0)}$. To partially resolve this ambiguity, we use a combination of 2d crossing symmetry, $\mathcal{W}$-algebra OPEs, and 6d selection rules to map operators between 2d and 6d:

- $W_6$: The conjecture of [1] associates the $S_6$ realization of $\mathcal{D}[6, 0]$ to $W_6$, by construction. Explicit $\mathcal{W}$-algebra OPEs computed with OPEdefs show that $W_6$ is only exchanged in $\langle S_4 S_4 S_4 S_4 \rangle|_\rho = \langle W_4 W_4 W_4 W_4 \rangle$, so that $\mathcal{D}[6, 0]$ superprimaries appearing in lower-rank correlators are associated with composite quasi-primaries.

- $[T T]^{(2)}$: We will see that comparing $\langle S_2 S_2 S_p S_p \rangle|_\rho$ with an $SL(2, \mathbb{R})$ block expansion written in terms of $\mathcal{W}$-algebra data fixes $[T T]^{(\ell + 2)}$ to be realized by $\mathcal{B}[2, 0]_\ell$.

- $[T W_3]^{(1)}$: $\mathcal{W}_N$ generators obey a $\mathbb{Z}_2$ symmetry $W_i \to (-1)^i W_i$ that prevents $[T W_3]^{(1)}$ from being exchanged in any $S_i \times S_j|_\rho = W_i \times W_j$ OPE that admits $\mathcal{D}[6, 0]$ or $\mathcal{B}[2, 0]_0$. $\langle S_2 S_3 S_2 S_3 \rangle$ will show that $[T W_3]^{(1)}$ is associated to a superdescendant of $\mathcal{D}[3, 2]$.

---

[15]See [16, 62, 63] for explicit results of both methods for $\mathcal{F}^{(4d)}_{2222}, \mathcal{F}^{(4d)}_{2233}, \mathcal{F}^{(4d)}_{2323}$, and $\mathcal{F}^{(4d)}_{3333}$.

- $[TW_4]^{(0)}, [W_3 W_3]^{(0)}, [TTT]^{(0)}$: The relation $\mathcal{B}[2,0]_\ell|_\rho \sim [TT]^{(\ell+2)}$ implies that $[TW_4]^{(0)}$, $[W_3 W_3]^{(0)}$, and $[TTT]^{(0)}$ can only be associated with $\mathcal{D}[6,0]$. These operators have overlapping two-point functions and are therefore not clearly separable, beyond noting that $W_3$ is absent in theories of type $D_{N>3}$, and that $[TTT]^{(0)}$ is only exchanged in $W_4 \times W_4$ and is thus unrelated to the $\mathcal{D}[6,0]$ exchanges in $\langle S_2 S_4 S_2 S_4 \rangle$, $\langle S_3 S_3 S_2 S_4 \rangle$, and $\langle S_3 S_3 S_3 S_3 \rangle$, for example.

Having elaborated on this example, we emphasize that the goal of this section is to compute 6d CFT data for use in the numerical conformal bootstrap. Provided we can find explicit expressions for individual $\lambda_{k_1 k_2 \mathcal{X}} \lambda_{k_3 k_4 \mathcal{X}}$, the standard bootstrap algorithm would not benefit from a more refined 6d/2d operator map.

Applying our small-$\chi$ expansion matching strategy to $\mathcal{F}_{k_1 k_2 k_3 k_4}$ up to $\{k_i\} = 4$ will generate a series of sum rules relating 6d data and $\mathcal{W}$-algebra parameters. To disentangle these sum rules and find expressions for individual $\lambda_{k_1 k_2 \mathcal{X}} \lambda_{k_3 k_4 \mathcal{X}}$, we begin by studying low-rank correlators where certain multiplets $\mathcal{X}$ appear in isolation. Having fixed these expressions, we move on to higher-rank correlators and use our previous formulae to fix higher data, as we will demonstrate below. This method necessarily computes data case-by-case and we have confirmed the formulae for $\lambda_{k_1 k_2 \mathcal{B}[p,0]_\ell} \lambda_{k_3 k_4 \mathcal{B}[p,0]_\ell}$ reported below for at least 100 values of $\ell$.[16,17] Some expressions for $\lambda_{k_1 k_2 \mathcal{B}[p,0]_\ell} \lambda_{k_3 k_4 \mathcal{B}[p,0]_\ell}$ are unwieldy and unilluminating so we relegate them to the ancillary `Mathematica` notebook.

$\langle S_2 S_2 S_p S_p \rangle$: We begin with the data appearing in $\langle S_2 S_2 S_p S_p \rangle$, where we find that:

$$
\begin{aligned}
\lambda_{22\mathcal{D}[2,0]} \lambda_{pp\mathcal{D}[2,0]} &= \frac{2p}{c}, \\
\lambda_{22\mathcal{D}[4,0]} \lambda_{pp\mathcal{D}[4,0]} &= \frac{1}{3}\delta_{2,p} + \frac{p(10p+2)}{30c}, \\
\lambda_{22\mathcal{B}[2,0]_\ell} \lambda_{pp\mathcal{B}[2,0]_\ell} &= \delta_{2p} \frac{\sqrt{\pi}(\ell+1)(\ell+4)(\ell+5)\Gamma[\ell+9]}{2^{2(\ell+6)}9\Gamma\left[\ell+\frac{11}{2}\right]} \\
&\quad + \frac{1}{c}\frac{\sqrt{\pi}(\ell+1)(\ell+4)(\ell+5)p((\ell+4)(\ell+7)p+2)\Gamma[\ell+3]}{2^{2\ell+9}\Gamma\left[\ell+\frac{11}{2}\right]}.
\end{aligned}
\tag{58}
$$

As mentioned above, comparing $SL(2,\mathbb{R})$ block expansions of $\mathcal{F}_{22pp}$ in terms of 6d and $\mathcal{W}$-algebra data uniquely associates $\mathcal{D}[4,0]|_\rho \sim [TT]^{(0)}$[18] and $\mathcal{B}[2,0]_\ell|_\rho \sim [TT]^{(\ell+2)}$.[19] Higher-rank correlators $\mathcal{F}_{ppqq}$ will also exchange this tower of $[TT]^{(\ell+2)}$ quasi-primaries and we can fix all the associated $\mathcal{B}[2,0]_\ell$ contributions with the expression

$$
\lambda_{pp\mathcal{B}[2,0]_\ell} \lambda_{qq\mathcal{B}[2,0]_\ell} = \frac{\left(\lambda_{22\mathcal{B}[2,0]_\ell} \lambda_{pp\mathcal{B}[2,0]_\ell}\right)\left(\lambda_{22\mathcal{B}[2,0]_\ell} \lambda_{qq\mathcal{B}[2,0]_\ell}\right)}{\lambda_{22\mathcal{B}[2,0]_\ell} \lambda_{22\mathcal{B}[2,0]_\ell}}.
\tag{59}
$$

---

[16]This exceeds the range of exchanged spins included in the numerical implementations of the analogous multi-correlator bootstrap for 3d $\mathcal{N} = 8$ theories [15] and 4d $\mathcal{N} = 4$ theories [16].

[17]It would be nice to reproduce these results using the Lorentzian inversion formula. This would require explicitly computing every superblock appearing in each $\langle S_{k_1} S_{k_2} S_{k_3} S_{k_4} \rangle$ and applying the inversion formula of [67]. Alternatively, one could derive reduced superblocks and compute this data using the modified inversion formula of [13].

[18]We choose a basis in which $[TT]^{(0)}$ is orthogonal to $W_4$ so that $\mathcal{D}[4,0]$ never realizes $W_4$ in $\mathcal{F}_{22pp}$.

[19]Recall that we have set to zero all $\mathcal{B}[0,0]_\ell|_\rho \sim [W_1 W_1]^{(\ell+2)}$ contributions to $\mathcal{F}_{ppqq}$, as we are uninterested in decoupled free sectors.

$\langle S_2 S_p S_2 S_p \rangle$:   We find

$$
\begin{aligned}
\lambda^2_{2p\mathcal{D}[p,0]} &= \frac{2p}{c}, \\
\lambda^2_{2p\mathcal{D}[p+2,0]} &= \frac{2}{(p+1)(p+2)}\left(\delta_{2p} + 1 + \frac{1}{c}\frac{2p(8p-5)}{2p+1}\right), \\
\lambda^2_{2p\mathcal{D}[p,2]} &= \frac{2}{(p+1)(p+2)}\left(p + 1 + \frac{1}{c}(p-2)(3p-1)\right).
\end{aligned}
\tag{60}
$$

We could not deduce a complete formula for the coefficients $\lambda_{2p\mathcal{B}[p,0]_\ell}$ for even and odd $\ell$, however we include our results for $2 \leq p \leq 10$ in the attached `Mathematica` file. The form of our R-symmetry polynomials $Y^{a,b}_{mn}(\sigma, \tau)$ and the transformation property of OPE coefficients of two scalars and a spin-$\ell$ operator imply that $s$- and $u$-channel data obey

$$
\lambda_{k_1 k_2 \mathcal{O}_{\Delta,\ell,mn}}\lambda_{k_3 k_4 \mathcal{O}_{\Delta,\ell,mn}} = (-1)^\ell \lambda_{k_2 k_1 \mathcal{O}_{\Delta,\ell,mn}}\lambda_{k_3 k_4 \mathcal{O}_{\Delta,\ell,mn}},
\tag{61}
$$

and we will not separately write out data obtainable by this relation.

$SL(2,\mathbb{R})$ block expansions of $\mathcal{F}_{2p2p}$ uniquely associate $\mathcal{D}[p+2,0]|_\rho \sim [TW_p]^{(0)}$, $\mathcal{D}[p,2]|_\rho \sim [TW_p]^{(1)}$, and $\mathcal{B}[p,0]_\ell|_\rho \sim [TW_p]^{(\ell+2)}$. Higher-rank correlators $\mathcal{F}_{pqpq}$ will also exchange this tower of $[TW_p]^{(\ell+2)}$ quasi-primaries and we can fix all the associated $\mathcal{B}[p,0]_\ell$ contributions with appropriate ratios involving $\lambda_{2p\mathcal{B}[p,0]_\ell}\lambda_{pq\mathcal{B}[p,0]_\ell}$ and $\lambda^2_{2p\mathcal{B}[p,0]_\ell}$.

$\langle S_3 S_3 S_3 S_3 \rangle$:   Moving beyond the Virasoro sector, in $\mathcal{F}_{3333}$ we find

$$
\begin{aligned}
\lambda^2_{33\mathcal{D}[2,0]} &= \frac{9}{c}, \\
\lambda^2_{33\mathcal{D}[4,0]} &= \frac{1}{6}\left(C^4_{33}\right)^2 + \frac{768}{5c(5c+22)}, \\
\lambda^2_{33\mathcal{D}[6,0]} &= \frac{1}{10} + \frac{971}{308c} + \frac{1}{90}\left(C^4_{33}\right)^2 - \frac{3}{10}\lambda^2_{3,3,\mathcal{B}[2,0]_0} - \frac{128}{55(5c+22)}.
\end{aligned}
\tag{62}
$$

The correlator $\mathcal{F}_{3333}$ contains data involving two families of semi-short operators: $\mathcal{B}[2,0]_\ell$ and $\mathcal{B}[4,0]_\ell$. Following our strategy, we determine the first using (59), leaving us with the isolated expression for $\lambda^2_{33\mathcal{B}[4,0]_\ell}$ shown in the ancillary `Mathematica` file.

$\langle S_2 S_3 S_3 S_4 \rangle$:   We find

$$
\begin{aligned}
\lambda_{23\mathcal{D}[3,0]}\lambda_{34\mathcal{D}[3,0]} &= \frac{\sqrt{2}C^4_{33}}{\sqrt{c}}, \\
\lambda_{23\mathcal{D}[5,0]}\lambda_{34\mathcal{D}[5,0]} &= \frac{13\sqrt{2}C^4_{33}}{35\sqrt{c}}, \\
\lambda_{23\mathcal{D}[3,2]}\lambda_{34\mathcal{D}[3,2]} &= \frac{\sqrt{2}C^4_{33}}{5\sqrt{c}}.
\end{aligned}
\tag{63}
$$

The correlator $\mathcal{F}_{2334}$ only involves the $\mathcal{B}[3,0]_\ell$ semi-short operators, allowing us to obtain the expressions for $\lambda_{23\mathcal{B}[3,0]_\ell}\lambda_{34\mathcal{B}[3,0]_\ell}$ with even and odd $\ell$ shown in the ancillary `Mathematica` file.

$\langle S_3 S_3 S_2 S_4 \rangle$:   We find

$$
\lambda_{33\mathcal{D}[4,0]}\lambda_{24\mathcal{D}[4,0]} = \frac{2\sqrt{2}C^4_{33}}{\sqrt{c}},
$$

$$\lambda_{33\mathcal{D}[6,0]}\lambda_{24\mathcal{D}[6,0]} = \frac{11\sqrt{2}C_{33}^4}{15\sqrt{c}}, \tag{64}$$

$$\lambda_{33\mathcal{B}[4,0]_\ell}\lambda_{24\mathcal{B}[4,0]_\ell} = \frac{\sqrt{\pi}(\ell+1)(\ell(\ell+15)+48)\Gamma[\ell+10]}{2^{2\ell+15}(\ell+3)\Gamma\left[\ell+\frac{15}{2}\right]}\frac{\sqrt{2}C_{33}^4}{\sqrt{c}}.$$

$\langle S_2 S_4 S_4 S_4 \rangle$:   We find

$$\lambda_{24\mathcal{D}[4,0]}\lambda_{44\mathcal{D}[4,0]} = \frac{2\sqrt{2}C_{44}^4}{\sqrt{c}},$$

$$\lambda_{24\mathcal{D}[6,0]}\lambda_{44\mathcal{D}[6,0]} = \frac{14\sqrt{2}C_{44}^4}{15\sqrt{c}}, \tag{65}$$

$$\lambda_{24\mathcal{B}[4,0]_\ell}\lambda_{44\mathcal{B}[4,0]_\ell} = \frac{\sqrt{\pi}(\ell+1)(\ell(\ell+15)+47)\Gamma[\ell+10]}{2^{2\ell+13}3(\ell+3)\Gamma\left[\ell+\frac{15}{2}\right]}\frac{\sqrt{2}C_{44}^4}{\sqrt{c}}.$$

$\langle S_3 S_3 S_4 S_4 \rangle$:   We find

$$\lambda_{33\mathcal{D}[2,0]}\lambda_{44\mathcal{D}[2,0]} = \frac{12}{c},$$

$$\lambda_{33\mathcal{D}[4,0]}\lambda_{44\mathcal{D}[4,0]} = \frac{(c+3)\left(C_{33}^4\right)^2}{2(c+2)} - \frac{48}{5c}, \tag{66}$$

$$\lambda_{33\mathcal{D}[6,0]}\lambda_{44\mathcal{D}[6,0]} = \frac{(5c+11)\left(C_{33}^4\right)^2}{30(c+2)} - \frac{23}{35c} - \frac{3}{10}\lambda_{33\mathcal{B}[2,0]_0}\lambda_{44\mathcal{B}[2,0]_0}.$$

As with $\mathcal{F}_{3333}$, the correlator $\mathcal{F}_{3344}$ contains data involving $\mathcal{B}[2,0]_\ell$ and $\mathcal{B}[4,0]_\ell$ and we use (59) to isolate the expression for $\lambda_{33\mathcal{B}[4,0]_\ell}\lambda_{44\mathcal{B}[4,0]_\ell}$ shown in the ancillary `Mathematica` file.

$\langle S_3 S_4 S_3 S_4 \rangle$:   We find

$$\lambda_{34\mathcal{D}[3,0]}^2 = \frac{1}{3}\left(C_{33}^4\right)^2,$$

$$\lambda_{34\mathcal{D}[5,0]}^2 = \frac{(25c+71)\left(C_{33}^4\right)^2}{70(c+2)} - \frac{6}{c},$$

$$\lambda_{34\mathcal{D}[3,2]}^2 = \frac{\left(C_{33}^4\right)^2}{5(c+2)}, \tag{67}$$

$$\lambda_{34\mathcal{D}[7,0]}^2 = \frac{1}{35} + \frac{276}{385c} + \frac{(457c+1121)\left(C_{33}^4\right)^2}{6930(c+2)} - \frac{9}{35}\lambda_{34\mathcal{B}[3,0]_0}^2,$$

$$\lambda_{34\mathcal{D}[5,2]}^2 = \frac{6}{35} + \frac{78}{35c} - \frac{(13c+17)\left(C_{33}^4\right)^2}{455(c+2)} - \frac{3}{10}\lambda_{34\mathcal{B}[3,0]_1}^2.$$

The correlator $\mathcal{F}_{3434}$ contains data involving $\mathcal{B}[3,0]_\ell$ and $\mathcal{B}[5,0]_\ell$. Following our strategy, we determine the first using results from $\mathcal{F}_{2323}$ and $\mathcal{F}_{2334}$ by writing

$$\lambda_{34\mathcal{B}[3,0]_\ell}^2 = \frac{\left(\lambda_{23\mathcal{B}[3,0]_\ell}\lambda_{34\mathcal{B}[3,0]_\ell}\right)^2}{\lambda_{23\mathcal{B}[3,0]_\ell}\lambda_{23\mathcal{B}[3,0]_\ell}}, \tag{68}$$

and find expressions for $\lambda_{34\mathcal{B}[5,0]_\ell}^2$ shown in the ancillary `Mathematica` file.

$\langle S_4 S_4 S_4 S_4 \rangle$:   We find

$$
\begin{aligned}
\lambda^2_{44\mathcal{D}[2,0]} &= \frac{16}{c}\,, \\
\lambda^2_{44\mathcal{D}[4,0]} &= \frac{1}{6}\left(C^4_{44}\right)^2 + \frac{2352}{5c(5c+22)}\,, \\
\lambda^2_{44\mathcal{D}[6,0]} &= \frac{24(882-5c)}{(5c+22)(70c+29)} + \frac{7968}{c(5c+22)(70c+29)} + \frac{1}{9}\left(C^4_{44}\right)^2\,, \\
\lambda^2_{44\mathcal{D}[8,0]} &= \frac{1}{35} + \frac{1948}{363c} + \frac{493\left(C^4_{44}\right)^2}{20020} - \frac{124236}{7865(5c+22)} - \frac{1}{21}\lambda^2_{44\mathcal{B}[2,0]_2} - \frac{9}{35}\lambda^2_{44\mathcal{B}[4,0]_0}\,.
\end{aligned}
\tag{69}
$$

The correlator $\mathcal{F}_{4444}$ contains data involving three families of semi-short operators: $\mathcal{B}[2,0]_\ell$, $\mathcal{B}[4,0]_\ell$, and $\mathcal{B}[6,0]_\ell$. While can fix $\mathcal{B}[2,0]_\ell$ data using (59), we were unable to fix the $\mathcal{B}[4,0]_\ell$ contribution to yield an isolated expression for $\lambda^2_{44\mathcal{B}[6,0]_\ell}$. In particular, our previous strategy would suggest two inequivalent contributions to $\lambda^2_{44\mathcal{B}[4,0]_\ell}$ obtained from the pairs $\mathcal{F}_{2444}$ and $\mathcal{F}_{2424}$, as well as $\mathcal{F}_{3344}$ and $\mathcal{F}_{3333}$. We would write

$$
\lambda^2_{44\mathcal{B}[4,0]_\ell} \supset \frac{1}{\mathcal{Y}^{0,0}_{33}\mathcal{C}^{\mathcal{B}[4,0]_\ell,0,0}_{\mathcal{O}_{14+\ell,\ell+2,3\,3}}} \frac{\left(\mathcal{Y}^{-2,0}_{33}\mathcal{C}^{\mathcal{B}[4,0]_\ell,-4,0}_{\mathcal{O}_{14+\ell,\ell+2,3\,3}}\right)^2}{\mathcal{Y}^{-2,-2}_{33}\mathcal{C}^{\mathcal{B}[4,0]_\ell,-4,-4}_{\mathcal{O}_{14+\ell,\ell+2,3\,3}}} \frac{(\lambda_{24\mathcal{B}[4,0]_\ell}\lambda_{44\mathcal{B}[4,0]_\ell})^2}{\lambda_{24\mathcal{B}[4,0]_\ell}\lambda_{24\mathcal{B}[4,0]_\ell}}\,,
\tag{70}
$$

$$
\lambda^2_{44\mathcal{B}[4,0]_\ell} \supset \frac{(\lambda_{33\mathcal{B}[4,0]_\ell}\lambda_{44\mathcal{B}[4,0]_\ell})^2}{\lambda_{33\mathcal{B}[4,0]_\ell}\lambda_{33\mathcal{B}[4,0]_\ell}}\,,
\tag{71}
$$

where the additional factors are needed to convert between kinematical configurations. We were unable to devise an argument for how these two contributions should be correctly weighted, nor if they capture the full $\lambda^2_{44\mathcal{B}[4,0]_\ell}$.[20] This inability distinguish $\lambda^2_{44\mathcal{B}[4,0]_\ell}$ from $\lambda^2_{44\mathcal{B}[6,0]_{\ell-2}}$ is a manifestation of the aforementioned filtration problem and we require an additional principle to separate this sum rule. We offer the following preliminary comments:

- $SL(2,\mathbb{R})$ block expansions of lower-rank correlators associate $\mathcal{B}[4,0]_\ell$ with a combination of $[TW_4]^{(\ell+2)}$, $[W_3W_3]^{(\ell+2)}$, and $[TTT]^{(\ell+2)}$. In a large-$c_\mathfrak{g}$ expansion, $[TW_4]^{(\ell+2)}$ and $[W_3W_3]^{(\ell+2)}$ will contribute to $\lambda^2_{44\mathcal{B}[4,0]_\ell}$ at $O\left(c_\mathfrak{g}^{-2}\right)$ with $c_\mathfrak{g}$-dependence stemming from $C^4_{44}(\mathfrak{g})$. By contrast, the triple-trace Virasoro operator $[TTT]^{(\ell+2)}$ will contribute at $O\left(c_\mathfrak{g}^{-3}\right)$ and be independent of $C^4_{44}(\mathfrak{g})$.

- The leading contribution to $\mathcal{F}_{4444}$ in a large-$c_\mathfrak{g}$ expansion is $O(1)$ and comes from the identity and the reduction of the $:S_4\partial^\ell S_4:$ operator. These GFF contributions are captured by $\mathcal{D}[0,0]$, $\mathcal{D}[8,0]$, and $\mathcal{B}[6,0]_\ell$ and we can unambiguously associate any $O(1)$ contribution to the $\lambda^2_{44\mathcal{B}[4,0]_\ell+2}$, $\lambda^2_{44\mathcal{B}[6,0]_\ell}$ sum rule with $\lambda^2_{44\mathcal{B}[6,0]_\ell}$.

- If one conjectures that $\mathcal{B}[6,0]_\ell$ in $\mathcal{F}_{4444}$ is solely associated with the operator $[W_4W_4]^{(\ell+2)}$, one can compare $SL(2,\mathbb{R})$ block expansions to find the correct linear combination of $\mathcal{W}$-algebra data $C^{\mathcal{O}}_{44}G_{\mathcal{O}\mathcal{O}'}C^{\mathcal{O}'}_{44}$ for $\mathcal{O}=[TW_4]^{(\ell+4)}, [W_3W_3]^{(\ell+4)}, [TTT]^{(\ell+4)}$ to evaluate the $\mathcal{B}[4,0]_{\ell+2}$ contribution. While one might benefit from a convenient basis of primaries like the one used for quasi-primaries involving $:T^n:$ in [68], it is computationally expensive to generate enough norms and coefficients using `OPEdefs` to extrapolate a general $\ell$-dependent pattern.

We conclude by emphasizing that these sum rules are sufficient to solve the twisted multi-correlator problem involving all $\mathcal{F}_{k_1 k_2 k_3 k_4}(\chi)$ up to $\{k_i\} = 4$.

---

[20]In theories of type $D_{N>3}$, the only possible contribution from lower-rank correlators is given by (70).

### 5.3 Regge trajectories

The authors of [13] explored the interplay between $\mathcal{N} = (2, 0)$ supersymmetry and analyticity in spin [67] to organize CFT data appearing in $\langle S_2 S_2 S_2 S_2 \rangle$ into conformal Regge trajectories. Given the collection of CFT data we computed in the previous section, it will be interesting to confirm that the data appearing in more complicated correlators continues to be compatible with this structure. As a first step, we observe that the $\langle S_2 S_2 S_p S_p \rangle$ data in (58) are related along Regge trajectories for even $\ell$:

$$\lambda_{22\mathcal{D}[2,0]} \lambda_{pp\mathcal{D}[2,0]} = \lim_{\ell \to -4} \left[ \frac{\mathcal{Y}_{22}^{0,0}}{\mathcal{Y}_{11}^{0,0}} \mathcal{C}_{\mathcal{O}_{10+\ell,\ell+2,2\,2}}^{\mathcal{B}[2,0]_\ell,0,0} \lambda_{22\mathcal{B}[2,0]_\ell} \lambda_{pp\mathcal{B}[2,0]_\ell} \right], \tag{72}$$

$$\lambda_{22\mathcal{D}[4,0]} \lambda_{pp\mathcal{D}[4,0]} = \lim_{\ell \to -2} \left[ \mathcal{C}_{\mathcal{O}_{10+\ell,\ell+2,2\,2}}^{\mathcal{B}[2,0]_\ell,0,0} \lambda_{22\mathcal{B}[2,0]_\ell} \lambda_{pp\mathcal{B}[2,0]_\ell} \right], \tag{73}$$

where the additional factors are needed to correctly relate conformal block coefficients, as can be seen in (33). For the $\langle S_2 S_p S_2 S_p \rangle$ data we computed with $2 < p \leq 10$, we find

$$\lambda_{2p\mathcal{D}[p,0]}^2 = \lim_{\ell \to -4} \left[ \frac{\mathcal{Y}_{\frac{p}{2}+1\frac{p}{2}+1}^{2-p,2-p}}{\mathcal{Y}_{\frac{p}{2}\frac{p}{2}}^{2-p,2-p}} \mathcal{C}_{\mathcal{O}_{2p+6+\ell,\ell+2,\frac{p}{2}+1\frac{p}{2}+1}}^{\mathcal{B}[p,0]_\ell,2(2-p),2(2-p)} \lambda_{2p\mathcal{B}[p,0]_{\ell_{\text{even}}}}^2 \right], \tag{74}$$

$$\lambda_{2p\mathcal{D}[p+2,0]}^2 = \lim_{\ell \to -2} \left[ \mathcal{C}_{\mathcal{O}_{2p+6+\ell,\ell+2,\frac{p}{2}+1\frac{p}{2}+1}}^{\mathcal{B}[p,0]_\ell,2(2-p),2(2-p)} \lambda_{2p\mathcal{B}[p,0]_{\ell_{\text{even}}}}^2 \right], \tag{75}$$

$$\lambda_{2p\mathcal{D}[p,2]}^2 = \lim_{\ell \to -1} \left[ \frac{\mathcal{C}_{\mathcal{O}_{2p+6+\ell,\ell+2,\frac{p}{2}+1\frac{p}{2}+1}}^{\mathcal{B}[p,0]_\ell,2(2-p),2(2-p)}}{\mathcal{C}_{\mathcal{O}_{2p+5,1,\frac{p}{2}+1\frac{p}{2}+1}}^{\mathcal{D}[p,2],2(2-p),2(2-p)}} \lambda_{2p\mathcal{B}[p,0]_{\ell_{\text{odd}}}}^2 \right]. \tag{76}$$

The separate relations involving even and odd $\ell$ are consistent with the results in [67], wherein even and odd spin data are organized in two independent analytic functions. This separation persists for data $\lambda_{k_1 k_2 \mathcal{B}[p,0]_l} \lambda_{k_3 k_4 \mathcal{B}[p,0]_l}$ in more complicated mixed correlators, whereby data associated with 1/4-BPS $\mathcal{D}[p,2]$ operators lie at $\ell = -1$ limits of $\mathcal{B}[p,0]_{\ell_{\text{odd}}}$ data. In the correlators $\langle S_3 S_3 S_p S_q \rangle$, we find relations like

$$\lambda_{33\mathcal{D}[2,0]} \lambda_{pp\mathcal{D}[2,0]} = \lim_{\ell \to -4} \left[ \frac{\mathcal{Y}_{22}^{0,0}}{\mathcal{Y}_{11}^{0,0}} \mathcal{C}_{\mathcal{O}_{10+\ell,\ell+2,2\,2}}^{\mathcal{B}[2,0]_\ell,0,0} \lambda_{33\mathcal{B}[2,0]_\ell} \lambda_{pp\mathcal{B}[2,0]_\ell} \right], \tag{77}$$

$$\lambda_{33\mathcal{D}[4,0]} \lambda_{pq\mathcal{D}[4,0]} = \delta_{pq} \lim_{\ell \to -2} \left[ \mathcal{C}_{\mathcal{O}_{10+\ell,\ell+2,2\,2}}^{\mathcal{B}[2,0]_\ell,0,0} \lambda_{33\mathcal{B}[2,0]_\ell} \lambda_{pp\mathcal{B}[2,0]_\ell} \right]$$

$$+ \lim_{\ell \to -4} \left[ \frac{\mathcal{Y}_{33}^{0,p-q}}{\mathcal{Y}_{22}^{0,p-q}} \mathcal{C}_{\mathcal{O}_{14+\ell,\ell+2,3\,3}}^{\mathcal{B}[4,0]_\ell,0,2(p-q)} \lambda_{33\mathcal{B}[4,0]_\ell} \lambda_{pq\mathcal{B}[4,0]_\ell} \right], \tag{78}$$

$$\lambda_{33\mathcal{D}[6,0]} \lambda_{pq\mathcal{D}[6,0]} = \lim_{\ell \to -2} \left[ \mathcal{C}_{\mathcal{O}_{14+\ell,\ell+2,3\,3}}^{\mathcal{B}[4,0]_\ell,0,2(p-q)} \lambda_{33\mathcal{B}[4,0]_\ell} \lambda_{pq\mathcal{B}[4,0]_\ell} \right]. \tag{79}$$

Finally, the data appearing in $\langle S_4 S_4 S_4 S_4 \rangle$ obeys

$$\lambda_{44\mathcal{D}[2,0]}^2 = \lim_{\ell \to -4} \left[ \frac{\mathcal{Y}_{22}^{0,0}}{\mathcal{Y}_{11}^{0,0}} \mathcal{C}_{\mathcal{O}_{10+\ell,\ell+2,2\,2}}^{\mathcal{B}[2,0]_\ell,0,0} \lambda_{44\mathcal{B}[2,0]_\ell}^2 \right], \tag{80}$$

$$\lambda_{44\mathcal{D}[4,0]}^2 = \lim_{\ell \to -2} \left[ \mathcal{C}_{\mathcal{O}_{10+\ell,\ell+2,2\,2}}^{\mathcal{B}[2,0]_\ell,0,0} \lambda_{44\mathcal{B}[2,0]_\ell}^2 \right] + \lim_{\ell \to -4} \left[ \frac{\mathcal{Y}_{33}^{0,0}}{\mathcal{Y}_{22}^{0,0}} \mathcal{C}_{\mathcal{O}_{14+\ell,\ell+2,3\,3}}^{\mathcal{B}[4,0]_\ell,0,0} \lambda_{44\mathcal{B}[4,0]_\ell}^2 \right], \tag{81}$$

$$\lambda_{44\mathcal{D}[6,0]}^2 = \lim_{\ell \to -2} \left[ \mathcal{C}_{\mathcal{O}_{14+\ell,\ell+2,3\,3}}^{\mathcal{B}[4,0]_\ell,0,0} \lambda_{44\mathcal{B}[4,0]_\ell}^2 \right] + \lim_{\ell \to -4} \left[ \frac{\mathcal{Y}_{44}^{0,0}}{\mathcal{Y}_{33}^{0,0}} \mathcal{C}_{\mathcal{O}_{18+\ell,\ell+2,4\,4}}^{\mathcal{B}[6,0]_\ell,0,0} \lambda_{44\mathcal{B}[6,0]_\ell}^2 \right], \tag{82}$$

$$\lambda_{44\mathcal{D}[8,0]}^2 = \lim_{\ell \to -2} \left[ \mathcal{C}_{\mathcal{O}_{18+\ell,\ell+2,4\,4}}^{\mathcal{B}[6,0]_\ell,0,0} \lambda_{44\mathcal{B}[6,0]_\ell}^2 \right], \tag{83}$$

which holds even with our partial result for $\lambda^2_{44\mathcal{B}[4,0]_\ell}$.[21] While we will not list every relation here, we have confirmed that the data of every $\mathcal{D}[p,q]$ superprimary in our work can be found as a linear combination of negative $\ell$ limits of semi-short $\mathcal{B}[p,0]_\ell$ data.

These relations stem from analyticity in spin of the underlying $\mathcal{W}$-algebra.[22] $\mathcal{N} = (2,0)$ supersymmetry will require us to be more careful in studying analyticity in spin in 6d. It was demonstrated in [13] that supersymmetry softens the Regge behavior of the four-point function $\mathcal{G}_{2222}$. The R-symmetry channel functions $A^{k_{12},k_{34}}_{mn}$ in (5), encoding $\mathfrak{so}(5)_R$ representations with Dynkin label $[2n, 2(m-n)]$, displayed Regge behavior such that according to [67], they can be analytically continued down to spin $J > 1-m-n$.[23,24]

A crucial point is that the Regge trajectory for a superconformal primary is accompanied by trajectories for all of the superdescendant conformal primaries. Naively extrapolating superdescendant trajectories to the (possibly negative) bound $J > 1-m-n$ can give rise to spurious blocks and mismatched block coefficients. Such issues were resolved in [13] for $\langle S_2 S_2 S_2 S_2 \rangle$ using an interplay between superdescendant trajectories of *a priori* unrelated supermultiplets. For example, taking (72) for $p = 2$ and analytically continuing the full superblock $\mathfrak{G}^{0,0}_{\mathcal{B}[2,0]_\ell}$ to $\ell = -4 \leftrightarrow J = -2$ gives rise to numerous spurious blocks not appearing in $\mathfrak{G}^{0,0}_{\mathcal{D}[2,0]}$. Certain negative-$\ell$ blocks are accounted for using the 6d block identities

$$G^{0,0}_{\Delta,-2}(U,V) = 0, \qquad G^{0,0}_{\Delta,-4-\ell}(U,V) = \frac{\ell+1}{\ell+3} G^{0,0}_{\Delta,\ell}(U,V), \qquad \text{for} \quad \ell \geq -1. \tag{84}$$

However, spurious blocks were only fully canceled by the conjecture that the $\mathcal{D}[2,0]$ super-multiplet is obtained be a linear combination of the $\ell \to -4$ continuation of $\mathcal{B}[2,0]_\ell$ and the $(\Delta, \ell) \to (2, -2)$ continuation of the unprotected $\mathcal{L}[0,0]_{\Delta,\ell}$ multiplet. In our conventions, the statement of [13] is that

$$\lambda^2_{22\mathcal{D}[2,0]}\mathfrak{G}^{0,0}_{\mathcal{D}[2,0]} = \lim_{\ell \to -4}\left[ \frac{\mathcal{Y}^{0,0}_{22}}{\mathcal{Y}^{0,0}_{11}} \, \mathcal{C}^{\mathcal{B}[2,0]_\ell,0,0}_{\mathcal{O}_{10+\ell,\ell+2,2\,2}} \lambda^2_{22\mathcal{B}[2,0]_\ell} \, \mathfrak{G}^{0,0}_{\mathcal{B}[2,0]_\ell} \right] - 2\lambda^2_{22\mathcal{D}[2,0]} \lim_{\substack{\Delta \to 2 \\ \ell \to -2}}\left[ \mathfrak{G}^{0,0}_{\mathcal{L}[0,0]_{\Delta,\ell}} \right]. \tag{85}$$

In principle, this conjectures the following dynamical constraint on the unprotected datum $\lambda^2_{22\mathcal{L}[0,0]_{\Delta,\ell}}$:

$$\lim_{\substack{\Delta \to 2 \\ \ell \to -2}}\left[ \lambda^2_{22\mathcal{L}[0,0]_{\Delta,\ell}} \right] = -2\lambda^2_{22\mathcal{D}[2,0]} = -\frac{8}{c}. \tag{86}$$

Another issue they considered arose from (73) for $p = 2$, where they found

$$\lambda^2_{22\mathcal{D}[4,0]}\mathfrak{G}^{0,0}_{\mathcal{D}[4,0]} = \lim_{\ell \to -2}\left[ \mathcal{C}^{\mathcal{B}[2,0]_\ell,0,0}_{\mathcal{O}_{10+\ell,\ell+2,2\,2}} \lambda^2_{22\mathcal{B}[2,0]_\ell} \, \mathfrak{G}^{0,0}_{\mathcal{B}[2,0]_\ell} \right] - (\text{spurious blocks}). \tag{87}$$

---

[21]In particular, the sum rule relating $\lambda^2_{44\mathcal{B}[6,0]_\ell}$ and $\lambda^2_{44\mathcal{B}[4,0]_{\ell+2}}$ causes (82) to be satisfied automatically, regardless of the value of $\lambda^2_{44\mathcal{B}[4,0]_\ell}$. The identity (81) holds when $\lim_{\ell \to -4}\left[ \lambda^2_{44\mathcal{B}[4,0]_\ell} \right] = \frac{1}{2}\left( C^4_{44} \right)^2$, which is true for both double-trace contributions in (70) and (71).

[22]Importantly, the map $\rho$ in (24) associates a 6d operator with spin $\ell_{6d}$ with a 2d chiral operator with spin $\ell_{2d} = \ell_{6d} + 6$. This means that even when we take a 6d negative spin continuation down to $\ell_{6d} = -4$, we still remain within $\ell_{2d} > 1$ which was range of validity of the analyticity in spin proposed in [67].

[23]While we observe this pattern in Equation (3.1) of [13], we certainly do not claim it will be true for $A^{k_{12},k_{34}}_{mn}$ functions in a general correlator $\mathcal{G}_{k_1 k_2 k_3 k_4}$.

[24]Due to the organization by the authors of [13] of $\mathcal{G}_{2222}$ into reduced superblocks, we need to introduce an effective spin label $J(\mathcal{X})$, where $J(\mathcal{L}[0,0]) = \ell$ and $J(\mathcal{B}[2,0]) = \ell + 2$.

The spurious blocks in this limit amount to a $(\Delta, \ell) = (8, 0)$ block in the $[0, 0]$ R-symmetry channel, as well as $(7, -1)$ and $(9, -1)$ blocks in the $[0, 2]$ channel, none of which appear in $\mathfrak{G}^{0,0}_{\mathcal{D}[4,0]}$. It was noted that the limit $\ell = -2 \leftrightarrow J = 0$ falls outside of the domain of analyticity in spin for the channels $[0, 0]$ and $[0, 2]$ and the authors suggested that we can accept this non-analyticity and ignore any spurious blocks.[25]

While we have not endeavored to compute the full set of superblocks $\mathfrak{G}^{\Delta_{12}, \Delta_{34}}_{\mathcal{X}}$ needed to analyze Regge trajectories of all correlators studied in this work, the superblocks presented in the ancillary `Mathematica` file of [23] are sufficient to study the $\langle S_3 S_3 S_p S_p \rangle$ cases explicitly. Consider the analytic continuation to $\mathcal{D}[4, 0]$ in these more complicated correlators. In $\langle S_3 S_3 S_3 S_3 \rangle$, the datum $\lambda^2_{33\mathcal{D}[4,0]}$ in (62) splits neatly into single-trace and double-trace pieces. We find that these pieces are separately the limits of two different semi-short multiplets:

$$\lambda^2_{33\mathcal{D}[4,0]}\big|_{:S_2 S_2:} = \frac{768}{5c(5c+22)} = \lim_{\ell \to -2} \left[ C^{\mathcal{B}[2,0]_\ell, 0, 0}_{\mathcal{O}_{10+\ell, \ell+2, 2\,2}} \lambda^2_{33\mathcal{B}[2,0]_\ell} \right], \tag{88}$$

$$\lambda^2_{33\mathcal{D}[4,0]}\big|_{S_4} = \frac{1}{6}\left(C^4_{33}\right)^2 = \lim_{\ell \to -4} \left[ \frac{\mathcal{Y}^{0,0}_{33}}{\mathcal{Y}^{0,0}_{22}} C^{\mathcal{B}[4,0]_\ell, 0, 0}_{\mathcal{O}_{14+\ell, \ell+2, 3\,3}} \lambda^2_{33\mathcal{B}[4,0]_\ell} \right]. \tag{89}$$

The $: S_2 S_2 :$ contribution is the same kind that appears in $\langle S_2 S_2 S_2 S_2 \rangle$ and can be treated analogously to (87), i.e.

$$\frac{768}{5c(5c+22)} \mathfrak{G}^{0,0}_{\mathcal{D}[4,0]} = \lim_{\ell \to -2} \left[ C^{\mathcal{B}[2,0]_\ell, 0, 0}_{\mathcal{O}_{10+\ell, \ell+2, 2\,2}} \lambda^2_{33\mathcal{B}[2,0]_\ell} \mathfrak{G}^{0,0}_{\mathcal{B}[2,0]_\ell} \right] - (\text{spurious blocks}). \tag{90}$$

The $S_4$ contribution is a novelty of studying the $S_3 \times S_3$ OPE and requires a different resolution. Indeed, we find that we achieve the same cancellation of (most) spurious blocks with the combination

$$\frac{1}{6}\left(C^4_{33}\right)^2 \mathfrak{G}^{0,0}_{\mathcal{D}[4,0]} = \lim_{\ell \to -4} \left[ \frac{\mathcal{Y}^{0,0}_{33}}{\mathcal{Y}^{0,0}_{22}} C^{\mathcal{B}[4,0]_\ell, 0, 0}_{\mathcal{O}_{14+\ell, \ell+2, 3\,3}} \lambda^2_{33\mathcal{B}[4,0]_\ell} \mathfrak{G}^{0,0}_{\mathcal{B}[4,0]_\ell} \right] - \frac{1}{2}\left(C^4_{33}\right)^2 \lim_{\substack{\Delta \to 6 \\ \ell \to -2}} \left[ \mathfrak{G}^{0,0}_{\mathcal{L}[2,0]_{\Delta, \ell}} \right]$$
$$- (\text{spurious blocks}), \tag{91}$$

where now the unprotected $\mathcal{L}[2,0]_{\Delta, \ell}$ multiplet is needed to cancel unwanted operators and we have the constraint that

$$\lim_{\substack{\Delta \to 6 \\ \ell \to -2}} \lambda^2_{33\mathcal{L}[2,0]_{\Delta, \ell}} = -3\lambda^2_{33\mathcal{D}[4,0]}\big|_{S_4} = -\frac{1}{2}\left(C^4_{33}\right)^2. \tag{92}$$

With this combination, we are left with precisely $\mathfrak{G}^{0,0}_{\mathcal{D}[4,0]}$ plus the same set of spurious blocks with identical coefficients in the $[0, 0]$ and $[0, 2]$ channels as in (87). To argue that we can again ignore these spurious blocks would require knowledge of the Regge behavior of $\langle S_3 S_3 S_3 S_3 \rangle$.[26] We postpone further speculation and note that the superblocks in $\langle S_3 S_3 S_3 S_3 \rangle$

---

[25] [13] alternatively proposed that an unprotected $\mathcal{L}$ multiplet with $(\Delta, \ell) = (4, -4)$ could cancel these spurious blocks. We were unable to achieve this with $\mathcal{L}[0,0]_{4,-4}$ without introducing several other unwanted blocks and will therefore not pursue this resolution.

[26] While it is plausible that the $\langle S_3 S_3 S_3 S_3 \rangle$ channel functions $A^{0,0}_{mn}$ continue to exhibit analyticity in spin for $J > 1 - m - n$ for the cases $m + n \leq 4$ that appeared in $\langle S_2 S_2 S_2 S_2 \rangle$, the new superblocks arising in $\mathcal{D}[3,0] \times \mathcal{D}[3,0]$ may modify this. To conduct the analogous study as in [13], we would need to analyze the three reduced correlator functions $a_{0,0}(z, \bar{z})$, $a_{1,0}(z, \bar{z})$, and $a_{0,1}(z, \bar{z})$ that arise in $\langle S_3 S_3 S_3 S_3 \rangle$ [20] and their behavior on the second sheet around the branch cut starting at $\bar{z} = 1$. The (unknown) reduced block decomposition of $a_{i,j}(z, \bar{z})$ would yield $J(\mathcal{B}[4,0]_\ell)$ and hence, the extent to which we can neglect certain spurious blocks that appear when considering negative-$\ell$ limits to $\mathcal{D}[4,0]$ and $\mathcal{D}[6,0]$ in (91) and (97).

are sufficient to make the analogous observation for $\langle S_3 S_3 S_p S_p \rangle$. When $p = 4$ for instance,

$$\lambda_{33\mathcal{D}[4,0]}\lambda_{44\mathcal{D}[4,0]}\Big|_{:S_2 S_2:} = \frac{1344}{5c(5c+22)} = \lim_{\ell \to -2}\Big[ \mathcal{C}^{\mathcal{B}[2,0]_\ell,0,0}_{\mathcal{O}_{10+\ell,\ell+2,2\,2}} \lambda_{33\mathcal{B}[2,0]_\ell}\lambda_{44\mathcal{B}[2,0]_\ell} \Big],$$

$$\lambda_{33\mathcal{D}[4,0]}\lambda_{44\mathcal{D}[4,0]}\Big|_{S_4} = \frac{(c+3)\big(C_{33}^4\big)^2}{2(c+2)} - \frac{48(c+10)}{c(5c+22)} \tag{93}$$

$$= \lim_{\ell \to -4}\Bigg[ \frac{\mathcal{Y}_{33}^{0,0}}{\mathcal{Y}_{22}^{0,0}} \mathcal{C}^{\mathcal{B}[4,0]_\ell,0,0}_{\mathcal{O}_{14+\ell,\ell+2,3\,3}} \lambda_{33\mathcal{B}[4,0]_\ell}\lambda_{44\mathcal{B}[4,0]_\ell} \Bigg].$$

Then similarly we have

$$\frac{1344}{5c(5c+22)}\mathfrak{G}^{0,0}_{\mathcal{D}[4,0]} = \lim_{\ell \to -2}\Big[ \mathcal{C}^{\mathcal{B}[2,0]_\ell,0,0}_{\mathcal{O}_{10+\ell,\ell+2,2\,2}} \lambda_{33\mathcal{B}[2,0]_\ell}\lambda_{44\mathcal{B}[2,0]_\ell} \mathfrak{G}^{0,0}_{\mathcal{B}[2,0]_\ell} \Big]$$
$$- \text{(spurious blocks)}, \tag{94}$$

and

$$\left( \frac{(c+3)\big(C_{33}^4\big)^2}{2(c+2)} - \frac{48(c+10)}{c(5c+22)} \right)\mathfrak{G}^{0,0}_{\mathcal{D}[4,0]} = \lim_{\ell \to -4}\Bigg[ \frac{\mathcal{Y}_{33}^{0,0}}{\mathcal{Y}_{22}^{0,0}} \mathcal{C}^{\mathcal{B}[4,0]_\ell,0,0}_{\mathcal{O}_{14+\ell,\ell+2,3\,3}} \lambda_{33\mathcal{B}[4,0]_\ell}\lambda_{44\mathcal{B}[4,0]_\ell} \mathfrak{G}^{0,0}_{\mathcal{B}[4,0]_\ell} \Bigg]$$
$$- 3\left( \frac{(c+3)\big(C_{33}^4\big)^2}{2(c+2)} - \frac{48(c+10)}{c(5c+22)} \right) \lim_{\substack{\Delta \to 6 \\ \ell \to -2}}\Big[ \mathfrak{G}^{0,0}_{\mathcal{L}[2,0]_{\Delta,\ell}} \Big]$$
$$- \text{(spurious blocks)}, \tag{95}$$

which would give the constraint that

$$\lim_{\substack{\Delta \to 6 \\ \ell \to -2}} \lambda_{33\mathcal{L}[2,0]_{\Delta,\ell}}\lambda_{44\mathcal{L}[2,0]_{\Delta,\ell}} = -3\lambda_{33\mathcal{D}[4,0]}\lambda_{44\mathcal{D}[4,0]}\Big|_{S_4}$$
$$= -3\left( \frac{(c+3)\big(C_{33}^4\big)^2}{2(c+2)} - \frac{48(c+10)}{c(5c+22)} \right). \tag{96}$$

The $\mathcal{D}[6,0]$ contribution to $\langle S_3 S_3 S_p S_p \rangle$ involves $: S_3 S_3 :$ and other multi-traces[27] and obeys

$$\lambda_{33\mathcal{D}[6,0]}\lambda_{pp\mathcal{D}[6,0]}\mathfrak{G}^{0,0}_{\mathcal{D}[6,0]} = \lim_{\ell \to -2}\Big[ \mathcal{C}^{\mathcal{B}[4,0]_\ell,0,0}_{\mathcal{O}_{14+\ell,\ell+2,3\,3}} \lambda_{33\mathcal{B}[4,0]_\ell}\lambda_{pp\mathcal{B}[4,0]_\ell} \mathfrak{G}^{0,0}_{\mathcal{B}[4,0]_\ell} \Big]$$
$$- \text{(spurious blocks)}, \tag{97}$$

where the spurious blocks are $\ell = -1$ blocks in the $[0,2]$ and $[2,2]$ channels and a $\Delta = 12$ scalar in the $[0,4]$ channel. This is analogous to the situation with the $: S_2 S_2 :$ superprimary of $\mathcal{D}[4,0]$ in $\langle S_2 S_2 S_p S_p \rangle$. Again, whether or not we can neglect these spurious blocks and attribute them to a breakdown of analyticity in spin will depend on the Regge behavior of $\mathcal{G}_{33pp}$ and the effective spin label $J(\mathcal{B}[4,0]_\ell)$, i.e. how $\mathcal{B}[4,0]_\ell$ blocks are organized in the reduced superblock decomposition of $\langle S_3 S_3 S_p S_p \rangle$ alluded to in Footnote 26. Considering this discussion and the relation (82), in $\langle S_4 S_4 S_p S_p \rangle$ we might expect an additional relation involving $\mathcal{B}[6,0]_{-4}$ and $\mathcal{L}[4,0]_{10,-2}$ which would capture the single-trace $S_6$ exchange that only appears in $S_4 \times S_4$ and beyond. We postpone further discussion and alternative resolutions to the spurious block problems to a more comprehensive study of $\langle S_3 S_3 S_p S_p \rangle$ and $\langle S_4 S_4 S_p S_p \rangle$ in future work.

---

[27]In particular, when $p = 3$ we only have $: S_3 S_3 :$ while for $p > 3$ we have a linear combination of $: S_3 S_3 :$, $: S_2 S_4 :$, and $: S_2 S_2 S_2 :$.

# 6 Consistency with perturbative M-theory

6d $(2,0)$ theories of type $\mathfrak{g} = A_{N-1}$ were conjectured to be holographically dual to M-theory on $AdS_7 \times S^4$ in one of the earliest examples of the AdS/CFT correspondence [5]. The duality was extended to relate $(2,0)$ theories of type $\mathfrak{g} = D_N$ to the M-theory orbifold $AdS_7 \times S^4/\mathbb{Z}_2$ in [6, 7]. It will be convenient to collectively refer to these M-theory duals as $AdS_7 \times S_4/\mathbb{Z}_{\mathfrak{o}}$, where the "orbifold factor" $\mathfrak{o}$ equals 1 for $\mathfrak{g} = A_{N-1}$ and 2 for $\mathfrak{g} = D_N$.

It has been a long-standing goal in holography to constrain the full S-matrix of supergravitons in $D$-dimensional string or M-theory on $\mathbb{R}^D$ using $CFT_d$ correlators, their duality to scattering on $AdS_{d+1} \times \mathcal{M}_{D-d-1}$, and an appropriate $D$-dimensional flat space limit thereof [69–74]. This program has been initiated for M-theory on $\mathbb{R}^{11}$ from the perspective of 3d $\mathcal{N} = 8$ ABJM theory in [17, 75–81] and for 6d $(2, 0)$ theory in [17, 23, 36, 82]. The M-theory supergraviton S-matrix $\mathcal{A}$ can be organized in small-momentum expansion in terms of the 11d Planck length $\ell_{11}$ as

$$
\begin{aligned}
\mathcal{A}(s,t) = \ell_{11}^9 \mathcal{A}_R \big( 1 &+ \ell_{11}^6 f_{R^4} + \ell_{11}^8 f_{D^2 R^4} + \ell_{11}^9 f_{R|R} + \ell_{11}^{10} f_{D^4 R^4} + \ell_{11}^{12} f_{D^6 R^4} \\
&+ \ell_{11}^{14} f_{D^8 R^4} + \ell_{11}^{15} f_{R|R^4} + \ell_{11}^{16} f_{D^{10} R^4} + \ell_{11}^{18} \big( f_{R|R|R} + f_{D^{12} R^4} \big) + \dots \big),
\end{aligned}
\tag{98}
$$

where $R$ denotes contributions from 11d supergravity, $f_{\mathcal{O}} = \frac{\mathcal{A}_{\mathcal{O}}}{\mathcal{A}_R}$ are polynomials in the 11d Mandelstam variables $s, t$ with *a priori* unfixed coefficients, and the expressions "$\mathcal{O}_1 | \dots | \mathcal{O}_{n+1}$" denote an $n$-loop diagram with vertices $\mathcal{O}_i$. Among these terms, only the tree-level higher-derivative terms up to $D^6 R^4$ are believed to be protected by supersymmetry and known via duality to type II string theory. Further, the $D^2 R^4$ term is redundant while $D^4 R^4$ can be shown to vanish via string theory. As a proof of concept of this M-theory bootstrap, [36] used the $A_{N-1}$ theory and its reduction to $\mathcal{W}_N$ to demonstrate consistency between M-theory EFT and CFT data appearing in $\langle S_k S_k S_k S_k \rangle$ and exactly recovered the $R^4$ term from $\langle S_3 S_3 S_3 S_3 \rangle$. The authors of [23] also recovered 1-loop terms involving $R$ and $R^4$ vertices using the analytic bootstrap.[28] In this section, we will demonstrate that the conjecture of [1] continues to be consistent with M-theory EFT expectations for any four-point correlator $\langle S_{k_1} S_{k_2} S_{k_3} S_{k_4} \rangle$ in 6d $(2,0)$ theories of type $\mathfrak{g} = A_{N-1}, D_N$.

Compactification on $AdS_7 \times S^4/\mathbb{Z}_{\mathfrak{o}}$ introduces powers of $L_{S^4} = \frac{1}{2} L_{AdS_7}$ into our expansion parameter, which we wish to recast in terms holographic 6d $(2,0)$ theory quantities. The holographic dictionary [5] and the anomaly arguments of [84] identify

$$
\left( \frac{L_{AdS_7}}{\ell_{11}} \right)^9 = 16 \mathfrak{o} c_{\mathfrak{g}} + O\left( c_{\mathfrak{g}}^0 \right).
\tag{99}
$$

The author of [84] demonstrated that at large $N$, the central charge $c_{\mathfrak{g}}$ can itself be organized in terms of $\mathfrak{o}$ as[29]

$$
c_{\mathfrak{g}} = (2\mathfrak{o})^2 N^3 + O(N^2),
\tag{100}
$$

---

[28]By contrast, more progress can made be from the ABJM (i.e. M2 brane) perspective thanks to its Lagrangian formulation [83]. Using constraints from supersymmetric localization, [75] and [76] recovered the $R^4$ and (lack of a) $D^4 R^4$ term, while [81] numerically recovers the $D^6 R^4$ term and offers a prediction for the unprotected $D^8 R^4$ term. The authors of [79] recovered 1-loop terms involving $R$ and $R^4$ vertices using the analytic bootstrap, fixing contact terms using localization.

[29]$O(N^2)$ terms are absent in $A_{N-1}$ theories, however holography allowed the $O(N^1)$ term to be computed from $R^4$ corrections in [85] and the $O(N^0)$ term from 1-loop 11d supergravity calculations in [86, 87].

allowing us to arrange our perturbative M-theory expansion (98) in terms of $c_{\mathfrak{g}}$ and $\mathfrak{o}$.[30] Compactification and holography allow us to recast the 11d supergraviton amplitude (98) into the 6d (2,0) correlator $\mathcal{G}_{k_1 k_2 k_3 k_4} \equiv \mathcal{G}_{\{k_i\}}$, which we expand as

$$
\begin{aligned}
\mathcal{G}_{\{k_i\}} = \mathcal{G}^{(\text{GFF})}_{\{k_i\}} + \frac{1}{c_{\mathfrak{g}}}\mathcal{G}^{(R)}_{\{k_i\}} + \frac{1}{\mathfrak{o}^{2/3}c_{\mathfrak{g}}^{5/3}}\mathcal{G}^{(R^4)}_{\{k_i\}} + \frac{1}{\mathfrak{o}^{4/3}c_{\mathfrak{g}}^{7/3}}\mathcal{G}^{(D^6 R^4)}_{\{k_i\}} + \sum_{n=8}^{\infty}\frac{1}{\mathfrak{o}^{\frac{2n+6}{9}}c_{\mathfrak{g}}^{\frac{2n+15}{9}}}\mathcal{G}^{(D^{2n}R^4)}_{\{k_i\}} \\
+ \sum_{\{l\}}\frac{F_{\{l\},\mathfrak{o}}}{c_{\mathfrak{g}}^{\#_{\{l\}}}}\mathcal{G}^{(\text{loops}_{\{l\}})}_{\{k_i\}} .
\end{aligned}
\tag{102}
$$

The first line contains the generalized free field contribution and tree-level contributions from supergravity and the higher-derivative corrections $D^{2n}R^4$ with $n \neq 1,2$ allowed by M-theory. The second line contains all loop corrections constructed from the vertices in the first line, where the exponent $\#_{\{l\}}$ can be determined by dimensional analysis. In analogy to the ABJM discussion in [81], integrating out the $S^4/\mathbb{Z}_{\mathfrak{o}}$ yields a factor $\text{Vol}(S^4/\mathbb{Z}_{\mathfrak{o}}) \sim \mathfrak{o}^{-1}$ in the $AdS_7$ effective action which cancels the $\mathfrak{o}$-dependence in the supergravity terms and staggers the power-law $\mathfrak{o}$-dependence in the remaining tree-level higher-derivative terms $D^{2n}R^4$. The loop terms require a sum over Kaluza-Klein modes on $S^4/\mathbb{Z}_{\mathfrak{o}}$ and therefore have a more complicated $\mathfrak{o}$-dependence encoded in $F_{\{l\},\mathfrak{o}}$.

The authors of [36] argued that CFT data associated with $\mathcal{W}$-algebra type multiplets $\mathcal{X}$ appearing in (22) should only encode properties of the protected terms in M-theory, i.e. $R$, $R^4$, and $D^6 R^4$. Their data should therefore truncate to include just

$$
\begin{aligned}
\lambda_{k_1 k_2 \mathcal{X}} \lambda_{k_3 k_4 \mathcal{X}} = \lambda^{(\text{GFF})}_{k_1 k_2 \mathcal{X}} \lambda^{(\text{GFF})}_{k_3 k_4 \mathcal{X}} + \frac{1}{c_{\mathfrak{g}}}\lambda^{(R)}_{k_1 k_2 \mathcal{X}}\lambda^{(R)}_{k_3 k_4 \mathcal{X}} + \frac{1}{\mathfrak{o}^{2/3}c_{\mathfrak{g}}^{5/3}}\lambda^{(R^4)}_{k_1 k_2 \mathcal{X}}\lambda^{(R^4)}_{k_3 k_4 \mathcal{X}} \\
+ \frac{1}{\mathfrak{o}^{4/3}c_{\mathfrak{g}}^{7/3}}\lambda^{(D^6 R^4)}_{k_1 k_2 \mathcal{X}}\lambda^{(D^6 R^4)}_{k_3 k_4 \mathcal{X}} + \sum_{\substack{i,j,k \\ i+j+k>1}}\frac{F_{i,j,k,\mathfrak{o}}}{c_{\mathfrak{g}}^{i+\frac{5}{3}j+\frac{7}{3}k}}\lambda^{(\text{loops}_{i,j,k})}_{k_1 k_2 \mathcal{X}}\lambda^{(\text{loops}_{i,j,k})}_{k_3 k_4 \mathcal{X}},
\end{aligned}
\tag{103}
$$

where the last term is a sum over contributions from $(i+j+k-1)$-loop diagrams involving $i$, $j$, and $k$ vertices of type $R$, $R^4$, and $D^6 R^4$, respectively.

The first check of our $\mathcal{W}$-algebra results is at $O(c_{\mathfrak{g}}^{-1})$, where 11d supergravity calculations provide an independent expression for $\lambda^{(R)}_{k_1 k_2 \mathcal{D}[p,0]}\lambda^{(R)}_{k_3 k_4 \mathcal{D}[p,0]}$ for $p < k_i + k_j$. Recall from Section 2.2 that these non-extremal couplings are dual to cubic couplings of single-particle states in supergravity. As summarized in [17], we have that

$$
\frac{1}{c_{\mathfrak{g}}}\lambda^{(R)}_{k_1 k_2 \mathcal{D}[p,0]}\lambda^{(R)}_{k_3 k_4 \mathcal{D}[p,0]} = \left(\mathcal{Y}^{k_{12},k_{34}}_{\frac{p}{2}\frac{p}{2}}\right)^{-1} g_{k_1 k_2 p}\, g_{k_3 k_4 p},
\tag{104}
$$

where $\mathcal{Y}^{k_{12},k_{34}}_{\frac{p}{2}\frac{p}{2}}$ was defined in (27). The quantities $g_{k_i k_j p}$ are cubic couplings between Kaluza-Klein scalars $\phi_{k_i}$ in the $AdS_7$ effective action. They were computed in [88,89] for $AdS_7 \times S^4$, which using (100) implies that for $AdS_7 \times S^4/\mathbb{Z}_{\mathfrak{o}}$ we have

$$
g_{k_1 k_2 k_3} = \frac{2^{k_1+k_2+k_3-1}}{(\pi^3 c_{\mathfrak{g}})^{\frac{1}{2}}}\Gamma\left[\frac{k_1+k_2+k_3}{2}\right]\frac{\Gamma\left[\frac{k_2+k_3-k_1+1}{2}\right]\Gamma\left[\frac{k_3+k_1-k_2+1}{2}\right]\Gamma\left[\frac{k_1+k_2-k_3+1}{2}\right]}{\sqrt{\Gamma[2k_1-1]\Gamma[2k_2-1]\Gamma[2k_3-1]}}.
\tag{105}
$$

---

[30]This is analogous to the situation in ABJM theory, i.e. the 3d $\mathcal{N} = 8$ $U(N)_k \times U(N)_{-k}$ Chern-Simons theory which, for fixed $k = 1, 2$, is the worldvolume theory of M2 branes on a $\mathbb{C}^4/\mathbb{Z}_k$ singularity. At large $N$ this is dual to $AdS_4 \times S^7/\mathbb{Z}_k$ and the dictionary and sphere free energy computations give [75]

$$
\left(\frac{L_{AdS_4}}{\ell_{11}}\right)^9 = \frac{3\pi}{2^{11}}k c_{N,k} + O(c_{N,k}^0), \qquad c_{N,k} = \frac{2^6}{3\pi}(2k)^{1/2}N^{3/2} + O(k^{-1/2}N^{1/2}).
\tag{101}
$$

Our results for all (non-extremal) $\lambda_{k_1 k_2 \mathcal{D}[p,0]}\lambda_{k_3 k_4 \mathcal{D}[p,0]}$ in both $A_{N-1}$ and $D_N$ theories up to $\{k_i\} = 4$ match (104) exactly. This agreement was demonstrated in [1] for $\mathfrak{g} = A_{N-1}$ for any $\lambda^{(R)}_{k_1 k_2 \mathcal{D}[p,0]}$ using a closed-form expression for $\mathcal{W}_N$ structure constants at large $N$ derived from [90]. We are unaware of analogous results for truncations of $\mathcal{W}_\infty^e$ which we need for a general matching when $\mathfrak{g} = D_N$.

To move beyond 11d supergravity contributions, we need to study the non-trivial $c_\mathfrak{g}^{-1}$ expansion in the structure constants of $\mathcal{W}_\mathfrak{g}$. As a result of the VOA bootstrap analysis, we only need to study the expansion of $C_{44}^4(\mathfrak{g})^2$. By expressing (42) and (47) in terms of $c_\mathfrak{g}$, we have the following large-$c_\mathfrak{g}$ expansions[31]

$$
\begin{aligned}
C_{44}^4(A_{N-1})^2 &= \frac{144}{5c_A} - \frac{2^{\frac{1}{3}}1080}{c_A^{5/3}} - \frac{88848}{25c_A^2} + \frac{2^{\frac{2}{3}}1080}{c_A^{7/3}} + \frac{2^{\frac{1}{3}}25128}{c_A^{8/3}} + \frac{2^{\frac{2}{3}}790632}{c_A^{10/3}} + O(c_A^{-3}), \\
C_{44}^4(D_N)^2 &= \frac{144}{5c_D} - \frac{2^{\frac{2}{3}}540}{c_D^{5/3}} - \frac{61848}{25c_D^2} + \frac{2^{\frac{1}{3}}540}{c_D^{7/3}} + \frac{2^{\frac{2}{3}}11214}{c_D^{8/3}} + \frac{2^{\frac{1}{3}}280566}{c_D^{10/3}} + O(c_D^{-3}),
\end{aligned}
\tag{106}
$$

which we can unify by writing

$$
C_{44}^4(\mathfrak{g})^2 = \frac{1}{c_\mathfrak{g}}\frac{144}{5} - \frac{1}{\mathfrak{o}^{2/3}c_\mathfrak{g}^{5/3}}2^{\frac{1}{3}}1080 + \frac{1}{\mathfrak{o}^{4/3}c_\mathfrak{g}^{7/3}}2^{\frac{2}{3}}1080 + \sum_{\substack{i,j,k \\ i+j+k>1}} \frac{F_{i,j,k,\mathfrak{o}}}{c_\mathfrak{g}^{i+\frac{5}{3}j+\frac{7}{3}k}}\left(C_{44}^{4\,(\text{loops}_{i,j,k})}\right)^2. \tag{107}
$$

This is precisely the structure (103) that we expect from M-theory arguments. In particular, there are no $\mathfrak{o}^{-8/9}c_\mathfrak{g}^{-17/9}$ or $\mathfrak{o}^{-10/9}c_\mathfrak{g}^{-19/9}$ terms, consistent with absence of $D^2 R^4$ and $D^4 R^4$ terms in M-theory EFT. In addition, the $A_{N-1}$ and $D_N$ theory values differ by orbifold factors in the expected way, which was not obvious from the constructions of $\mathcal{W}_N$ and $\mathcal{W}_{D_N}$ in Section 4. Finally, the $c_\mathfrak{g}^{-(2n+7)/9}$ terms with $n > 7$ do not have power-law scalings in $\mathfrak{o}$, so we associate them with loops built from $R, R^4$, and $D^6 R^4$ vertices, as opposed to unprotected tree-level $D^{2m}R^4$ vertices with $m \geq 8$ which are not expected to be captured by $\mathcal{W}_\mathfrak{g}$. Some correlators depend instead on the combinations $(C_{33}^4)^2$, $c_A^{-1/2}C_{33}^4$, or $c_\mathfrak{g}^{-1/2}C_{44}^4(\mathfrak{g})$ and one can check that these combinations exhibit the same $c_\mathfrak{g}^{-1}$ expansion structure as (107).

The discussion in Section 5.1 implied that every twisted correlator $\mathcal{F}_{k_1 k_2 k_3 k_4}$ for any $\{k_i\}$ can be expressed in terms of just $c_\mathfrak{g}$ and $C_{44}^4(\mathfrak{g})$, so all CFT data in this protected subsector will inherit the scaling (103), other than for $\langle S_2 S_2 S_p S_p \rangle$ and its crossed channels where data is $c_\mathfrak{g}^{-1}$-exact. In particular, semi-short coefficients $\lambda_{k_1 k_2 \mathcal{B}[p,0]_\ell}\lambda_{k_3 k_4 \mathcal{B}[p,0]_\ell}$ receives $R^4$ and $D^6 R^4$ higher-derivative corrections. This comes in contrast to the computation of other unknown coefficients and anomalous dimensions in [23,91,92], where long $\lambda_{22\mathcal{L}[0,0]_{\Delta,\ell}}^2$ data only receives $R^4$ corrections for $\ell = 0$ and semi-short $\lambda_{22\mathcal{B}[0,2]_\ell}^2$ data receive no $R^4$ correction at all. Similarly, semi-short data in the ABJM theory does not receive $R^4$ corrections [75,81]. These restrictions come from the strategy proposed by [93], wherein unknown CFT data is determined exactly by demanding crossing symmetry on a truncated superconformal block expansion order-by-order in $c^{-1}$. By virtue of VOA associativity, the $\mathcal{W}_\mathfrak{g}$ algebra sector of 6d (2,0) theories satisfies crossing symmetry automatically at every order in $c_\mathfrak{g}^{-1}$ and all data in this sector is corrected by $R^4$, $D^6 R^4$, and loops thereof thanks to their dependence on $C_{44}^4(\mathfrak{g})$.

By incorporating the second family of holographic 6d (2,0) theories, i.e. those of type $D_N$, our results complete a non-trivial consistency check between the 6d (2,0)/$\mathcal{W}$-algebra conjecture, holography, and the protected structures in perturbative M-theory.

---

[31]We note that at higher orders in $1/c_D$, the $C_{44}^4(D_N)^2$ expansion develops complex factors (cubic roots of unity) which depend on the choice of root $N(c_D)$ chosen when inverting (46). The quantity $C_{44}^4(D_N)^2$ is manifestly real and the complex factors are artifacts of expanding (47) in large $c_D$ instead of $N$. With these considerations, one can unambiguously extract the real coefficient of any desired order in $1/c_D$.

# 7 Discussion and outlook

The main result of our paper was to extract explicit formulae for 6d (2,0) CFT data $\lambda_{k_1 k_2 \mathcal{X}} \lambda_{k_3 k_4 \mathcal{X}}$ from the conjectured correspondence to $\mathcal{W}_{\mathfrak{g}}$ algebras. Along the way, we outlined technicalities involving the superblocks of mixed correlators in (2,0) theories and their chiral algebra twist. We also elucidated finite-$N$ aspects of 6d (2,0) $\mathcal{W}_{\mathfrak{g}}$ algebras beyond the Virasoro algebra, including $\mathcal{W}_{D_N}$, which have not previously appeared in the 6d (2,0) literature. Using our explicit 6d data, we commented on generalizations of the Regge trajectory discussion in [13] for the correlators $\langle S_3 S_3 S_p S_p \rangle$. Finally, we demonstrated the consistency of the $\mathcal{W}_{\mathfrak{g}}$ algebra conjecture for $\mathfrak{g} = A_{N-1}, D_N$ with protected higher-derivative corrections in M-theory.

The most obvious future direction is to use our CFT data in a numerical multi-correlator bootstrap program analogous to the studies of 3d $\mathcal{N} = 8$ theories in [15] and 4d $\mathcal{N} = 4$ theories in [16]. This could involve either comparing bounds on $\mathcal{W}$-algebra data with our explicit formulae or inputting these formulae to obtain sharper bounds on data not captured by the $\mathcal{W}$-algebra. If one pursued the former strategy and the exact data saturated the numerical bounds, one could use the extremal functional method of [94] to extract the spectrum and approximations of OPE coefficients for all remaining CFT data. It would also be nice to devise a way to impose the Regge trajectory constraints that relate certain protected single-trace $\mathcal{D}$ data to negative spin limits of unprotected $\mathcal{L}$ data.

One obstacle is that the standard numerical bootstrap used in [11, 12] converges substantially more slowly in 6d than in lower dimensions. While the semidefinite program solver SDBP [10] has been upgraded [95] since the implementation in [12], it would be interesting to attack the 6d (2,0) bootstrap using alternative numerical strategies. For example, the iterative application of the Lorentzian inversion formula in [13] could be applied to higher-$k_i$ four-point functions where the $\mathcal{W}$-algebra seed data is more complicated and theory-dependent. One could also incorporate the dispersion relation of [96] in intermediate steps in the iteration, especially now that a mixed correlator dispersion kernel has been derived in [97], which is needed given the form of reduced superblocks of $\langle S_2 S_2 S_p S_p \rangle$.

The reinforcement learning approach proposed in [98, 99] was applied to 6d (2,0) theories in [14]. Their method was subsequently improved and complemented by another non-convex optimization algorithm in [100]. The implementation for a 1d defect CFT bootstrap problem in [101] demonstrated these strategies' ability to reproduce rigorous results in [102, 103], especially once supplied with known theoretical input. One difference is that the theoretical input we propose in 6d are a set of explicit OPE coefficients, whereas the input in the defect problem [101–103] were integrated correlator constraints obtained from localization.

Another direction is to examine 6d (2,0) correlation functions in the presence of defects, as has been pursued in e.g. [104–108].[32] Surface defects inserted orthogonal to the chiral algebra $(z, \bar{z})$ plane preserve the supercharge $\mathbf{Q}$ needed to define the cohomology, allowing us to extract exact defect CFT data from the VOA. It would be interesting to compute such data beyond leading order in $1/c_{\mathfrak{g}}$ (i.e. the 11d supergravity approximation) using our closed-form expressions for $C_{44}^4(\mathfrak{g})$. More generally, in 4d $\mathcal{N} = 2$ SCFTs the authors of [109] used an adapted topological descent procedure to construct and classify extended operators which live in the cohomology of twisted Schur supercharges found in the original chiral algebra construction of [35]. There should be no obstruction, at least in principle, to carry out the same construction and classification for the analogous extended operators in 6d (2,0) theories.

Beyond the conformal bootstrap, it would be interesting to understand how the holographic dual of $\mathcal{W}$-algebras arises from a top-down construction in 11d supergravity and more ambitiously, M-theory. The result should match the bottom-up picture given by minimal model

---

[32]We note that [108] appeared on the arXiv the day after the appearance of the first version of this work.

holography and its even-spin counterpart [58, 59, 110, 111], where $\mathcal{W}_N$ and $\mathcal{W}_{D_N}$ appear naturally as boundary duals. A top-down supergravity derivation would involve a localization procedure using the cohomology of a certain Killing spinor, i.e. the bulk realization of the supercharge **Q**. The analogous question was addressed in the duality between 4d $\mathcal{N} = 4$ SCFTs and type IIB string theory on $AdS_5 \times S^5$ in [112] where as a proof of concept, the authors performed this localization on a simplified setup involving 5d super Yang-Mills on a non-dynamical $AdS_5$ background. A more recent top-down approach was provided by twisted holography, where the authors of [113] conjectured that the chiral algebra sector of holography between 4d $\mathcal{N} = 4$ SCFTs and type IIB strings on $AdS_5 \times S^5$ is realized by a holographic construction in B-model topological string theory. Neither a convenient bulk toy model analogous to the one considered in [112], nor any practical notion of "topological M-theory" are available for deriving a twisted sector of M-theory on $AdS_7 \times S^4/\mathbb{Z}_0$ dual to 6d (2,0) $\mathcal{W}$-algebras. Thus, the first step in this direction will likely require a full 11d supergravity localization computation.

## Acknowledgments

We are very grateful to Shai Chester and Costis Papageorgakis for numerous helpful discussions about 6d (2,0) theories and especially to Costis Papageorgakis for comments on the draft.

**Funding information** This work was funded by a Science and Technology Facilities Council (STFC) studentship.

## A  6d (2,0) superconformal block conventions

While we will not work with block expansions of the full 6d four-point function (2), our analysis of the twisted correlator will depend on our conventions in 6d. In particular, we will follow the conventions of [23] by expressing 6d bosonic blocks found in [33, 34] as

$$
G_{\Delta,\ell}^{\Delta_{12},\Delta_{34}}(z,\bar{z}) \tag{A.1}
$$
$$
= \mathcal{F}_{0,0} - \frac{(\ell+3)}{\ell+1}\mathcal{F}_{-1,1} + \frac{2(\Delta-4)\Delta_{12}\Delta_{34}(\ell+3)}{(\Delta+\ell)(\Delta+\ell-2)(\Delta-\ell-4)(\Delta-\ell-6)}\mathcal{F}_{0,1}
$$
$$
+ \frac{(\Delta-4)(\ell+3)(\Delta-\Delta_{12}-\ell-4)(\Delta+\Delta_{12}-\ell-4)\big(\Delta+\Delta_{34}-\ell-4\big)\big(\Delta-\Delta_{34}-\ell-4\big)}{16(\Delta-2)(\ell+1)(\Delta-\ell-5)(\Delta-\ell-4)^2(\Delta-\ell-3)}\mathcal{F}_{0,2}
$$
$$
- \frac{(\Delta-4)(\Delta-\Delta_{12}+\ell)(\Delta+\Delta_{12}+\ell)\big(\Delta+\Delta_{34}+\ell\big)\big(\Delta-\Delta_{34}+\ell\big)}{16(\Delta-2)(\Delta+\ell-1)(\Delta+\ell)^2(\Delta+\ell+1)}\mathcal{F}_{1,1}\,,
$$

where we define

$$
\mathcal{F}_{m,n}(z,\bar{z}) \tag{A.2}
$$
$$
\equiv \frac{(z\bar{z})^{\frac{\Delta-\ell}{2}}}{(z-\bar{z})^3}\left((-z)^\ell z^{m+3}\bar{z}^n \,_2F_1\left(\frac{\Delta+\ell-\Delta_{12}}{2}+m, \frac{\Delta+\ell+\Delta_{34}}{2}+m, \Delta+\ell+2m; z\right)\right.
$$
$$
\left. \times \,_2F_1\left(\frac{\Delta-\ell-\Delta_{12}}{2}-3+n, \frac{\Delta-\ell+\Delta_{34}}{2}-3+n, \Delta-\ell-6+2n; \bar{z}\right) - (z \leftrightarrow \bar{z})\right).
$$

The $\mathfrak{so}(5)_R$ harmonic functions $Y_{mn}^{a,b}(\sigma,\tau)$ are computed using the scheme described in [17, 19]. In particular, $Y_{mn}^{a,b}(\sigma,\tau)$ are polynomials

$$
Y_{mn}^{a,b}(\sigma,\tau) = \sum_{i=0}^{m}\sum_{j=0}^{m-i} c_{i,j}\sigma^i \tau^j\,, \tag{A.3}
$$

where we solve for the coefficients $c_{i,j}$ using the $SO(\mathtt{d})$ Casimir eigenfunction equation

$$L^2 Y_{mn}^{a,b}(\sigma,\tau) = -2C_{mn}^{a,b} Y_{mn}^{a,b}(\sigma,\tau), \tag{A.4}$$

with quadratic Casimir operator[33]

$$L^2 = 2\mathcal{D}_{\mathtt{d}}^{a,b} - (a+b)(a+b+\mathtt{d}-2), \tag{A.5}$$

in terms of operators

$$\mathcal{D}_{\mathtt{d}}^{a,b} = \mathcal{D}_{\mathtt{d}} + (1-\sigma-\tau)(a\partial_\tau + b\partial_\sigma) - 2a\sigma\partial_\sigma - 2b\tau\partial_\tau, \tag{A.6}$$

$$\mathcal{D}_{\mathtt{d}} = (1-\sigma-\tau)(\partial_\sigma\sigma\partial_\sigma + \partial_\tau\tau\partial_\tau) - 4\sigma\tau\partial_\sigma\partial_\tau - (\mathtt{d}-2)(\sigma\partial_\sigma + \tau\partial_\tau), \tag{A.7}$$

and eigenvalues

$$C_{mn}^{a,b} = \left(m + \frac{a+b}{2}\right)\left(m + \frac{a+b}{2} + \mathtt{d} - 3\right) + \left(n + \frac{a+b}{2}\right)\left(n + \frac{a+b}{2} + 1\right). \tag{A.8}$$

We normalize $Y_{mn}^{a,b}(\sigma,\tau)$ such that the $\sigma^m$ term has coefficient $c_{m,0} = 1$. For consistency with crossing symmetry, we must normalize solutions corresponding to $1 \leftrightarrow 2$ crossed channels such that the $\tau^m$ term has coefficient $c_{0,m} = 1$.

Twisting a 6d superblock (11) in the manner described in Section 3.2 will leave us with a global $SL(2,\mathbb{R})$ block of the form

$$g_{\Delta,l}^{\Delta_{12},\Delta_{34}}(\chi) = (-1)^l \chi^{\frac{\Delta+l}{2}} {}_2F_1\left(\frac{\Delta+l}{2} - \frac{\Delta_{12}}{2}, \frac{\Delta+l}{2} + \frac{\Delta_{34}}{2}, \Delta + l, \chi\right). \tag{A.9}$$

# B $\;\mathcal{W}$-algebra structure constant relations from the VOA bootstrap

## B.1 $\;\mathcal{W}_{A_{N-1}} = \mathcal{W}_N$

Here, we list the explicit $\mathcal{W}_N$ structure constants that appear in the 6d four-point functions considered in this work. The complete set of data determined by Jacobi identities $(W_i W_j W_k)$ up to $i + j + k \leq 12$ can be found in the ancilliary Mathematica file. The latter corrects a small number of typos in [46]. From the $(W_3 W_3 W_4)$ Jacobi identities, we have

$$C_{34}^3 = \frac{C_{33}^4 C_{44}^0}{C_{33}^0}, \qquad C_{35}^4 = \frac{5(c+7)(5c+22)}{(c+2)(7c+114)} \frac{\left(C_{33}^4\right)^2 C_{44}^0}{C_{33}^0 C_{34}^5} - \frac{60}{c} \frac{C_{33}^0}{C_{34}^5},$$

$$C_{44}^4 = \frac{3(c+3)}{c+2} \frac{C_{33}^4 C_{44}^0}{C_{33}^0} - \frac{288(c+10)}{c(5c+22)} \frac{C_{33}^0}{C_{33}^4}, \qquad C_{44}^6 = \frac{4}{5} \frac{C_{34}^5 C_{35}^6}{C_{33}^4}, \tag{B.1}$$

$$C_{44}^{[33]^{(0)}} = \frac{4}{5} \frac{C_{34}^5 C_{35}^{[33]^{(0)}}}{C_{33}^4} + \frac{30(5c+22)}{(c+2)(7c+114)} \frac{C_{44}^0}{C_{33}^0}.$$

The normalization of $W_6$ is indirectly captured by the structure constant $C_{35}^6$. The shift symmetry of $W_6$ with respect to the spin-6 composite primary $[W_3 W_3]^{(0)}$ is fixed by $C_{35}^{[33]^{(0)}}$. However, these choices drop out of the correlators we will consider. From the $(W_3 W_4 W_5)$ Jacobi identities, we have

$$C_{55}^0 = \frac{5(c+7)(5c+22)}{(c+2)(7c+114)} \frac{\left(C_{33}^4\right)^2 \left(C_{44}^0\right)^2}{C_{33}^0 \left(C_{34}^5\right)^2} - \frac{60}{c} \frac{C_{33}^0 C_{44}^0}{\left(C_{34}^5\right)^2}, \tag{B.2}$$

---

[33]We correct a typo in the definition of $L^2$ in [17] as compared with [19].

where normalizing $W_5$ through $C_{55}^0$ fixes $\left(C_{34}^5\right)^2$. For brevity, we will not list structure constants involving $T$ and composites thereof. These are determined by Virasoro symmetry and are automatically computed using the `MakeOPE` command in `OPEconf`.

We should point out that, in contrast to our case-by-case approach using `OPEdefs`, other methods conjecture more general expressions for $\mathcal{W}_N$ structure constants $C_{ij}^k$. For example, [90] provides a closed form expression for all structure constants in the classical $\mathcal{W}_\infty[\mu]$ algebra in the primary basis. When $\mu = N$, [1] argue these should match those of $\mathcal{W}_N$ to leading order in the double scaling limit $N \to \infty, c \to \infty, \frac{c}{4N^3} \to 1$. Alternatively, the bounded nonlinearity of the quadratic basis allowed [46] to conjecture a closed-form expression for all $\mathcal{W}_N$ structure constants in this basis.

## B.2 $\quad \mathcal{W}_{D_N}$

The complete set of $\mathcal{W}_{D_N}$ associativity relations determined from Jacobi identities $(W_i W_j W_k)$ up to $i+j+k \leq 16$ can be found in Appendix A of [47], where no normalizations and orthogonality conditions were imposed. The only relation we need in our work is derived from the Jacobi identities for $(W_4 W_4 W_4)$ and $(W_4 W_4 W_6)$ and reads

$$C_{66}^0 = \frac{4(5c+22)}{9(c+24)} \frac{C_{44}^0 \left(C_{44}^4\right)^2}{\left(C_{44}^6\right)^2} - \frac{96((c-172)c+196)}{c(2c-1)(7c+68)} \frac{\left(C_{44}^0\right)^2}{\left(C_{44}^6\right)^2}. \tag{B.3}$$

In analogy to (B.2), this relation fixes $\left(C_{44}^6\right)^2$ in terms of $\left(C_{44}^4\right)^2$ upon choosing normalizations for $W_4$ and $W_6$ through $C_{44}^0$ and $C_{66}^0$. For the reader's convenience, we include the full list of relations from $(W_i W_j W_k)$ up to $i + j + k \leq 14$ in the ancillary `Mathematica` file.

## B.3 $\quad \mathcal{W}_{E_n}$ with $n = \{6, 7, 8\}$

For the curious reader, we list some low-lying associativity constraints on the structure constants of $\mathcal{W}$-algebras associated with the 6d (2,0) theories of exceptional type $\mathfrak{g} = E_6, E_7, E_8$. According to (35), these theories and their $\mathcal{W}$-algebras have central charges

$$c(E_6) = 3750, \qquad c(E_7) = 9583, \qquad c(E_8) = 29768. \tag{B.4}$$

We begin with $E_6$, where the algebra $\mathcal{W}_{E_6}$ should be generated by currents

$$\{T, W_5, W_6, W_8, W_9, W_{12}\}.$$

After encoding these and the most general ansatz for their OPEs in `OPEconf`, the Jacobi identites for $(W_5 W_5 W_5)$ and $(W_5 W_5 W_6)$ yield

$$C_{55}^6 = \frac{30404351116253971248}{106383311368274257291175} \frac{C_{55}^0}{C_{66}^6}, \qquad C_{55}^8 = \frac{10293716465979085952}{44556149344718466714675} \frac{C_{55}^0 C_{66}^8}{\left(C_{66}^6\right)^2},$$

$$C_{56}^5 = \frac{11914010550}{1985085221} C_{66}^6, \qquad C_{58}^5 = \frac{376908281010555590033536725}{1166128012510074836999206} \frac{\left(C_{66}^6\right)^2}{C_{66}^8},$$

$$C_{59}^6 = \frac{1431429014008559307}{165914847495201472} \frac{C_{59}^8 C_{66}^6}{C_{66}^8}, \qquad C_{56}^9 = \frac{2235484260987977728}{203269888317569937\,6625} \frac{C_{55}^0 C_{66}^8}{C_{59}^8 C_{66}^6}, \tag{B.5}$$

$$C_{68}^8 = -\frac{10511936419336}{11642524821165} C_{66}^6, \qquad C_{68}^6 = \frac{45247875130269756381327932069}{158593409701370177831892016\,0} \frac{\left(C_{66}^6\right)^2}{C_{66}^8}.$$

Next we consider $E_7$, where the algebra $\mathcal{W}_{E_7}$ has generating currents

$$\{T, W_6, W_8, W_{10}, W_{12}, W_{14}, W_{18}\},$$

and the Jacobi identities from $(W_6, W_6, W_6)$ yield

$$
\begin{aligned}
C_{66}^0 &= \frac{27458051496202588389994853950 69}{1076044622523216113178984000}\left(C_{66}^6\right)^2 \\
&\quad + \frac{4657671734711644921 3903}{11595458323303983340 5}C_{66}^8 C_{68}^6 - \frac{1488074036649087834 31}{169839797234930640}C_{66}^{10}C_{6,10}^6, \\
C_{6,10}^8 &= \frac{93595}{65542}\frac{C_{68}^8 C_{66}^8}{C_{66}^{10}} + \frac{209315479477}{128296806525}\frac{C_{66}^6 C_{66}^8}{C_{66}^{10}}, \\
C_{6,10}^{10} &= \frac{76680}{32771}\frac{C_{66}^8 C_{68}^{10}}{C_{66}^{10}} - \frac{63943397}{160621980}C_{66}^6.
\end{aligned}
\tag{B.6}
$$

Finally, we consider $E_8$, where $\mathcal{W}_{E_8}$ is generated by

$$\{T, W_8, W_{12}, W_{14}, W_{18}, W_{20}, W_{24}, W_{30}\}.$$

The $(W_8, W_8, W_8)$ Jacobi identities result in the constraints

$$
\begin{aligned}
C_{88}^0 &= \frac{6925404577403590259188816316880997984169199616}{880453459564356072197903068200081462193 75}\left(C_{88}^8\right)^2 \\
&\quad - \frac{1262945634053482878813101747538432}{2222872131372265271848189662 5}C_{88}^{12}C_{8,12}^8, \\
C_{8,14}^8 &= \frac{7746803635871960523136925916831176283}{2863420004156039641361020396578778 78}\frac{C_{88}^{12}C_{8,12}^8}{C_{88}^{14}} \\
&\quad - \frac{60834522926821294933520951315375214028073428156049772000 0}{18761310944847124249837667796830364279214337144596721016 7}\frac{\left(C_{88}^8\right)^2}{C_{88}^{14}}, \\
C_{8,14}^{12} &= \frac{29467}{46649}\frac{C_{8,12}^{12}C_{88}^{12}}{C_{88}^{14}} + \frac{2257533194753776342022264116 9}{2292354229372284259440554887 5}\frac{C_{88}^8 C_{88}^{12}}{C_{88}^{14}}, \\
C_{8,14}^{14} &= \frac{285831}{186596}\frac{C_{88}^{12}C_{8,12}^{14}}{C_{88}^{14}} - \frac{1759978756161731906554353 83}{503814116345556980096825250}C_{88}^8.
\end{aligned}
\tag{B.7}
$$

We are not aware of results that determine these coefficients in analogy to the minimal representation strategies used to determine $C_{33}^4$ in $\mathcal{W}_N$ and $C_{44}^4$ in $\mathcal{W}_{D_N}$. Nonetheless, these results demonstrate non-trivial relations between three-point coefficients of single-trace[34] operators in the exceptional 6d (2,0) theories that are accessible neither via holography nor by any convenient string or M-theory construction.

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
