# Peer review of "The $\mathcal{W}$-algebra bootstrap of 6d $\mathcal{N}=(2,0)$ theories"

_SciPost Physics, doi:SciPost Phys. 19, 074 (2025)_

## Round 2 · Referee Report · Anonymous (Referee 1) · 2025-7-11

Report
A basic conceptual question I have, is that the map between operators in 2d and 6d, at least as specified in the original paper 1404.1079, is ambiguous. This is bc protected operators in 6d are labeled by spin and dimension, while 2d operators are labelled just by their level.
For instance, the 6d operators B[4,0]_{j-2} and B[2,0]_j look identical in 2d for j>0, and similarly B[2,0]_0 and D[6]. in the 2d algebra, there are 6 level 6 operators (here d is a derivative and S_n is level n generator):
S_6
S_4S_2
S_2S_2S_2
S_3S_3
S_2 dd S_2
S_2 d S_3
and its not clear which of these is D[6] or B[2,0]_0 in general (except that obviously S_6 is D[6])
This problem (which is called the filtration problem) is already known in the more well studied 4d case, as described eg in 1612.08956. How exactly does the author propose to resolve this issue?
For some lowlying operators it should be possible to avoid this issue using 2d crossing. For instance, the 2d 3333 is fixed by crossing up to the infinite set of B[2,0]_l operators, and these can be fixed by considering the 2d 2222 and 2233, which are each totally fixed by crossing, since 2233 includes the ope coefficient 2 2 B[2,0]_l. Perhaps a similar indirect strategy could work for other correlators.
Requested changes
- clarify the filtration problem.
Recommendation
Ask for minor revision

Author: Mitchell Woolley on 2025-07-16 [id 5645]
(in reply to Report 1 on 2025-07-11)Thanks for this point - this was indeed an ambiguity that I encountered. As mentioned in the final remark, I fixed the contributions of low-lying operators using crossing among simpler correlators. For example, ppB[2,0]j can be fixed from 2222 and 22pp by the same logic, leaving only ppB[4,0]{j-2} and higher in the remaining correlators ppqq. An analogous strategy works for pqpq and similar. I will update Section 5.2 to make my crossing strategy more transparent.
The only instance where this ambiguity became a problem was in 4444, where crossing was not sufficient to distinuish between the B[4,0]j and B[6,0]{j-2} data, as declared at the end of Section 5.2. The aim of Section 5 was to compute data for comparison with (or as input for) the bootstrap of correlators up to 4444. Aside from 44B[4,0]j and 44B[6,0]{j-2}, I believe I correctly extracted individual data and the bootstrap would not benefit from a more detailed 6d-2d operator map. I will clarify in the paper how the latter ambiguity is a manifestation of the filtration problem in 6d, which I will also briefly describe in the paper.
Beyond the bootstrap, this is an interesting puzzle and below are some considerations that I used to relate 6d supermultiplets to 2d operators: 1) 2d block decomposition - One can always compare the twisted 6d block decomposition with a sl(2,R) block decomposition of a W_g correlator to derive sum rules relating 6d data to combinations of 2d OPE and 2-pt. coefficients. After fixing low-lying data using crossing among lower-rank correlators, one can find the right linear combination of 2d operator data that contribute to a 6d datum. In practice, it is computationally expensive (e.g. with OPEdefs) to evaluate enough 2d coefficients to extrapolate the l-dependent pattern of B-type data. In principle though, one can for instance compute the regular part of the S_p x S_q OPE to see which explicit 2d composite operators contribute to twisted D[p+q,0] and B-type blocks.
2) Holography/GFF - When D[p,0] x D[q,0] exchanges an extremal operator D[p+q,0], I exclude this from realizing the 6d "single trace" 1/2-BPS operator of rank p+q to ensure finiteness of the dual AdS_7 effective action at large c. Hence, I exclude the twisted D[p+q,0] block from realizing S_{p+q} in this correlator. The regular part of W-algebra OPEs confirm this interpretation for any c. Further, I interpret the O(c^0) piece of D[p+q,0] and B[p+q-2]j data as coming from the the tower of (short) double twist operators :D[p,0] d^j D[q,0]: (of appropriate so(5)_R projection) that survive in Generalized Free Field Theory (c->infinity). This does not fix the contributions of other multi-trace operators at subleading orders in 1/c however.
3) Analyticity in spin - As discussed in Section 5.3, taking certain negative-j limits of 6d B-type data gives rise to D-type data, which stems from analyticity in spin in 2d. There are two non-vanishing limits one can take: j -> -2 or j-> -4 and we find that D-type data D[p,0] can be expressed as a sum of j -> -4 limits of B[p,0]j and j -> -2 limits of B[p-2,0]j , and that these pieces exactly realize the 2d OPE coefficients coming from S_{p+q} and composite operator exchanges, respectively. This might give a principle for extracting the "single-trace" contribution of a given pqD[r,0] datum.
Regarding the 2d operators that could descend from D[6,0] and B[2,0]j and B[4,0]{j-2}: S_6 - I can use either holography or W-algebra OPEs (with a basis of composite quasi-primaries orthogonal to S_6) to argue it appears as D[6,0] in <4444> but not in any other correlator.
:S_2 d^j S_2: - I used my formula for B[2,0]j data appearing to <22pp> to fix the B[2,0]j contributions to higher pqrs. The known S_i x S_j OPEs imply that :S_2 d^0 S_2: contributes to D[4,0] while :S_2 d^{j+2} S_2: gives B[2,0]j.
:S_2 d^j S_3: - Using W_g OPEs, the Z_2 symmetry of W_g generators S_i -> (i)^{-1} S_i, and 6d selection rules, one can see that :S_2 d^j S_3: will never be exchanged in any twisted correlator that involves D[6,0] or B[4,0]{j-2}, so this candidate is ruled out. :S_2 d^j S_3: is instead associated with D[5,0], D[3,2], and B[3,0]{j-2}.
:S_2 d^j S_4:, :S_3 d^j S_3: and :S_2 S_2 d^j S_2: - Aside from the D_{N>3} theories (which are without S_3), these operators will generically have non-zero 2-point function and we can only single out the :S_2 d^{j+2} S_4: contribution 24B[4,0]j in 2424. I did not attempt to compute the 2-pt. coefficient of these operators in general. While this affects 3344, 2444, 3324, and 4444, crossing symmetry was enough to isolate expressions for pqB[4,0]j in all but 4444. :S_2 S_2 d^j S_2: quasi-primaries are only exchanged in S_4 x S_4 OPEs and beyond, and their only appearances in pq44 for p,q<4 come from 2-pt. coefficient overlaps with :S_2 d^j S_4:, :S_3 d^j S_3:, :S_2 d^{j+2} S_2:. My VOA calculations used canonical normalization and shift conventions for composite operators involving the Virasoro generator S_2, though perhaps non-canonical conventions would distinguish contributions to different twisted correlators more cleanly. This doesn't affect data of course, and one would need an additional principle that assigns :S_2 S_2 d^j S_2: to, say, B[4,0]{j-2} instead of B[6,0]j.
Finally, a comment on distinguishing 44B[4,0]j and 44B[6,0]{j-2}: A future direction is to reproduce my data using the Lorentzian inversion formula, as was pursued in 2105.13361 for 2222. A direct analogue of 2105.13361 requires a reduced superblock formalism for higher-rank correlators. Deriving reduced superblocks is the subject of my ongoing work, where I have preliminary results for 2p2p, 2pqq, and 33pp. The analogous calculation for 4444, combined with my closed-form twisted 4444 correlator, should give an independent result for 44B[4,0]j and 44B[6,0]{j-2}.

---

## Round 2 · Referee Report · Anonymous (Referee 2) · 2025-7-29

Strengths
1) The paper is clearly written and tackles a timely and relevant open problem in the literature. 2) The paper is logically structured and develops the arguments in a pedagogical and self-contained manner.
Weaknesses
1) Insuffient discussion of ambiguities in the approach. 2) The manuscript contains a few minor spelling and grammatical errors.
Report
The paper studies the 4pt bootstrap problem for 1/2-BPS operators in the 6d (2,0) theory of type $A_{N-1}$ and $D_N$. The strategy applied in this paper is to relate the protected part of these correlators to chiral algebra data. The author solves the meromorphic bootrap for all four-point functions involving operators of rank up to $k_i=4$ and obtains closed-form expressions for selected OPE coefficients. Various concistency checks are performed, including agreement with higher-derivative corrections in M-theory. The paper is well-written and logically structured.
As already pointed out by the first referee, the identification of 6d OPE data with chiral algebra data can be ambigious. I agree that this issue deserves clearer discussion. In particular, it would be helpful if the author could explicitly distinugish which OPE coefficients are unambiguously fixed by the chiral algebra, and which require additional input; either from a concrete understanding of the R-filtration or from independent 6d considerations. The reply of the author clarifies this to some extend, but a concise summary in the main text would be valuable.
No other major concerns were found. Below are a few minor comments that could improve the presentation: - It would be appropriate to briefly mention similar bootstrap strategies applied in 4d $\mathcal{N}=4$ SYM, where chiral algebra constraints have also been used to fix protected correlators. - Section 4 contains a few imprecise statement: (1) Further support for the Beem et al. conjecture comes from the $\Omega$-deformation of the holomorphic topological twist of the (2,0) theory, which more directly connects to the AGT correspondence where similar W-algebras appear. (2) $W_g$ algebras are obtained as the qDS reduction of the affine KM algebra not qDS reduction of a simple lie algebra. (3) The truncation of $W_{1+\infty}$ to $W_g$ is more subtle than currently stated: it requires specific values of the central charge c and an additional parameter \lambda, which is related to the roots introduced in equation (4.5). - In the final paragraph of the discussion, the author might wish to cite the more recent derivation of the $\mathcal{N}=4$ chiral algebra sector given by Costello and Gaiotto in their work on twisted holography.
Requested changes
1) Address the ambiguities and which additional data is needed to fix them. 2) Address the minor points raised above.
Recommendation
Ask for minor revision

---

## Round 3 · Referee Report · Anonymous (Referee 1) · 2025-8-11

Report

Please accept this paper.

Recommendation

Publish (meets expectations and criteria for this Journal)

---

## Round 3 · Referee Report · Anonymous (Referee 2) · 2025-8-11

Report

The author has adequately addressed all the concerns raised in the previous reports. I recommend this paper for publication.

Recommendation

Publish (meets expectations and criteria for this Journal)

---

## Round 3 · Author Response

Following the helpful reports from Referees 1 and 2 and the editorial recommendation, we have implemented minor revisions that clarify both the filtration ambiguity in the 6d/W-algebra conjecture and the strategy we used to obtain isolated expressions for 6d CFT data by applying crossing symmetry to a system of twisted correlators. We have elaborated on which data can be determined from crossing alone and point out explicitly the sole example where the filtration problem prevents us from obtaining individual expressions for 6d data. In addition, we corrected some imprecise statements and typos and added various additional references that relate the 6d/W-algebra correspondence to its analogue in 4d N=4 SCFTs and its holographic dual, as well as to the W symmetry appearing in the AGT correspondence.

---

## Round 3 · List of Changes

1) Clarifying the filtration problem in 6d: We added three pieces of text that identify this problem and its practical effect on our calculation. In particular, when defining the chiral algebra twist in Section 3.2, we point out how the cohomological reduction coarsens our organization of 6d operators by effectively removing a quantum number, and how the map between 6d and 2d operators is ambiguous, given the numerous ways one might realize a 6d operator of fixed weight with different 2d (non-)composite quasiprimaries. We briefly comment on the analogous problem in 4d N=2 theories. In Section 5.2, we describe the practical effect this has on our expansion matching strategy for computing 6d data. We also describe an example (the one in our response to Referee 1) of how additional 2d and 6d principles can be used to partially lift this ambiguity. Finally, we attribute our inability to distinguish B[4,0]j and B[6,0]{j-2} in <4444> to the filtration problem and offer some preliminary comments on interpreting the resulting sum rule.

2) Clarifying our data extraction strategy: We reorganized Section 5.2, first by being more upfront in identifying the filtration problem, but also detailing our strategy to determine individual CFT data contributing to higher-rank correlators by first pinning down their contributions to lower-rank correlators. The main examples of this are the determination of B[2,0]j from <22pp> and B[p,0]j from <2p2p>. Furthermore, we partitioned the results of each correlator into sub(sub)sections and in each case wrote sentences on how our strategy fixes low-lying semi-short data using lower-rank correlators, allowing for an isolated expression for the highest semi-short datum.

3) Reference to AGT: In Section 4 we included a paragraph briefly describing another important appearance of W symmetry in the context of 6d (2,0) theories, namely the AGT correspondence. We include references to subsequent constructions that give evidence to the compatibility of the 6d/W-algebra correspondence with AGT.

4) Comparison with chiral algebras of 4d N=4 SCFTs: In Section 5.1 we included a brief discussion of the analogous chiral algebra construction in 4d N=4 SCFTs. We discuss the relative utility of these constructions by pointing out that certain 4d N=4 chiral algebra results are equally obtainable by a free-field computation, whereas W-algebras are indispensable for interpreting the protected correlator in 6d. We also discuss differences in what these chiral algebras can determine holographically.

5) Twisted holography: We improve our chiral algebra holography discussion in Section 7 by briefly describing the twisted holography work by Gaiotto and Costello using topological string theory.

6) Errors: --- We corrected our statement of the parameters that W_{1+infinity} depends on, and the way that these are fixed upon a truncation to W_g. --- We corrected our statement to say that W_g are obtainable as the quantum Drinfel'd-Sokolov reduction of an affine Kac-Moody algebra \hat{g}. --- We corrected numerous mispellings and grammatical and mathematical typos.

---

## Editorial Decision

published